# A Complete Assessment of Dopamine Receptor- Ligand Interactions through Computational Methods

**DOI:** 10.3390/molecules24071196

**Published:** 2019-03-27

**Authors:** Beatriz Bueschbell, Carlos A. V. Barreto, António J. Preto, Anke C. Schiedel, Irina S. Moreira

**Affiliations:** 1PharmaCenter Bonn, Pharmaceutical Institute, Pharmaceutical Chemistry I, University of Bonn, D-53121 Bonn, Germany; bueschbell@uni-bonn.de (B.B.); schiedel@uni-bonn.de (A.C.S.); 2Center for Neuroscience and Cell Biology, UC- Biotech Parque Tecnológico de Cantanhede, Núcleo 04, Lote B, 3060-197 Cantanhede, Portugal; cbarreto@cnc.uc.pt (C.A.V.B.); martinsgomes.jose@gmail.com (A.J.P.); 3Institute for Interdisciplinary Research, University of Coimbra, 3004-531 Coimbra, Portugal

**Keywords:** dopamine receptors, molecular docking, molecular dynamics, receptor-ligand interactions

## Abstract

*Background*: Selectively targeting dopamine receptors (DRs) has been a persistent challenge in the last years for the development of new treatments to combat the large variety of diseases involving these receptors. Although, several drugs have been successfully brought to market, the subtype-specific binding mode on a molecular basis has not been fully elucidated. *Methods*: Homology modeling and molecular dynamics were applied to construct robust conformational models of all dopamine receptor subtypes (D_1_-like and D_2_-like). Fifteen structurally diverse ligands were docked. Contacts at the binding pocket were fully described in order to reveal new structural findings responsible for selective binding to DR subtypes. *Results*: Residues of the aromatic microdomain were shown to be responsible for the majority of ligand interactions established to all DRs. Hydrophobic contacts involved a huge network of conserved and non-conserved residues between three transmembrane domains (TMs), TM2-TM3-TM7. Hydrogen bonds were mostly mediated by the serine microdomain. TM1 and TM2 residues were main contributors for the coupling of large ligands. Some amino acid groups form electrostatic interactions of particular importance for D_1_R-like selective ligands binding. *Conclusions*: This in silico approach was successful in showing known receptor-ligand interactions as well as in determining unique combinations of interactions, which will support mutagenesis studies to improve the design of subtype-specific ligands.

## 1. Introduction

### 1.1. Dopamine Receptors

The dopaminergic system has been intensively studied over the past 75 years due to the (patho)physiological role in modulating cognitive and motor behaviour [1,2]. The importance of dopamine has dramatically emerged from being just an intermediate in the formation of noradrenaline to having a celebrity status as the most important mammalian neurotransmitter [3]. Dopamine binds to five distinct dopamine receptors (DRs; D_1–5_ Receptor), grouped into two classes —D_1_-like and D_2_-like receptors— that differ in their physiological effects and signal transduction. The D_1_-like receptors, D_1_R and D_5_R, are principally coupled to G_s_ proteins and enhance the activity of adenylyl-cyclase, whereas D_2_-like receptors, D_2–4_R, are primarily coupled to inhibitory G_i_ proteins and suppress the activity of adenylyl cyclases [1,4]. The DRs belong to the G Protein-Coupled Receptor family (GPCRs), the largest and most diverse protein family in humans with approximately 800 members [5,6]. GPCRs share a conserved overall fold of seven transmembrane helices (TMs) linked by three intracellular loops (ICLs) and three extracellular loops (ECLs). Around 30–40% of all available pharmacotherapeutics target this protein family [7].

Many severe neuropsychiatric and neurodegenerative disorders such as Tourette’s syndrome, schizophrenia, Parkinson’s disease and Huntington’s disease are believed to occur as a result of imbalances and alterations in dopamine signaling [8,9,10]. Moreover, also a wide array of psychiatric disorders such as hallucinations, paranoia, bipolar disorder, gambling, alcoholism, mania, depression, eating disorders, movement and hyperactivity disorders are linked to malfunctioning dopaminergic transmission [3,11,12,13]. The discovery of chlorpromazine, the first antipsychotic drug, in the 1950s, was the hallmark of the development of a collection of antipsychotics [14] which were later reported to commonly bind to the D_2_R subtype (with different affinity) [4,11]. The “first-generation/classical” antipsychotics came along with significant motor side effects such as tardive dyskinesia, extrapyramidal symptoms and related conditions. These problems were not eliminated in “second-generation/atypical” antipsychotics, and others such as weight gain and the “metabolic syndrome” also appeared [11,15,16,17]. It was then later discovered that these multiple clinical and adverse effects of several antipsychotics depended on the combination of occupied receptors from other systems such as cholinergic, histaminergic and serotoninergic receptors (but always including the D_2_R) resulting in non-selective profiles and therefore in an insufficient explanation of the mechanism of action [11,15]. Until today, no drug has yet been identified with antipsychotic action without a significant affinity for the D_2_R [15]. However, antipsychotics remain a necessary first-line treatment for the management of a variety of the already mentioned psychiatric disorders (Figure 1). In fact, it is difficult to directly target one of these diseases with one specific antipsychotic, since there are also numerous cases of non-responding patients to first-line or any antipsychotic treatment or which become resistant to treatment over time [11,18]. 

The search for DR subtype selective (foremost D_2_R-selective) therapeutics is an ongoing field of research [15] as current drugs have D_2_R/D_3_R-affinity or affinity for all DRs [19,20]. It has been proposed that substituted 4-phenylpiperazine compounds distinguish between D_3_R and D_2_R selectivity [21,22]. In addition, the aminotetraline derivative 7-OH-DPAT was identified as a selective D_3_R agonist [23,24], whereas it was shown that most D_4_R available therapeutics are not selective [22], with only one exception, haloperidole [25] (Figure 1). Also, finding D_1_-like DR targeting ligands, a more challenging aim [26,27], may improve antipsychotic treatment, as D_1_R was also shown to be relevant for modulating behaviour in health and disease (reviewed in O’Sullivan et al. [28]). So far, SKF38393 was the only selective agonist attained for the D_1_R, while D_5_R completely lacks a selective ligand [29,30]. SCH23390 was proposed as the only D_1_-like DR selective antagonist [31] (Figure 1). 

### 1.2. Computer-Aided Drug Design 

The strive for finding new and effective therapeutics led to a growing interest in the use of Computer-Aided Drug Design (CADD). Originally developed for high-throughput screening (HTS) of compound libraries, the use of CADD nowadays plays an important role in drug discovery [32]. The CADD pipeline can be classified into two general categories: structure-based and ligand-based, dependent on the available information about the topic of investigation [33]. A structure-based CADD is used when the target, e.g., a receptor, is known and compound libraries can be screened to find suitable structures for the target. In contrast, ligand-based CADD relies on known active and inactive compounds with their affinities in order to construct quantitative pharmacophore models and to perform virtual screening that is carried out target-independently [32]. Both CADD approaches are only fruitful if sufficient information is present. Structure-based approaches rely on the availability of the target crystal structure or homologs proteins whereas pharmacophore and other ligand-based methodologies rely on the existence of a sufficient number of ligands. For example, for GPCRs potentially involved in Parkinson’s disease, a variety of molecular docking studies were carried out using resolved crystal structures to which self-synthesized ligands were docked (reviewed in Lemos et al., [34]). *Vice versa*, inspection of known ligands was used to build pharmacophores or Quantitative Structure-Activity-Relationships (QSAR) to screen for new bioactive molecules (reviewed in Lemos et al., [34]). All in all, CADD is capable of addressing many challenges in hit-to-lead-development and is currently widely used in the pharmaceutical industry [34,35]. 

### 1.3. Aim 

Modeling GPCRs remains problematic due to the complex structure of these membrane proteins and the lack of structural information about the desired receptor to target. However, the recent boom of resolved X-ray crystallography structures leads to a more promising application of CADD approaches to this receptor. Herein, we used tools of structure-based CADD to investigate the receptor-ligand properties of all DR-subtypes with marketed DR therapeutics. In particular, we: (i) applied homology modeling by using the resolved X-ray crystallography structures of the dopamine receptors D_2_R, D_3_R and D_4_R (D_2_R bound to the atypical antipsychotic risperidone, PDBid: 6CM4 [36]; D_3_R bound to D_2_R/D_3_R-selective antagonist eticlopride, PDBid: 3PBL [37]; and D_4_R in complex with D_2_R/D_3_R-selective antagonist nemonapride, PDBid: 5WIU [38]) in order to provide models with structural ligand-free properties; (ii) performed Molecular Dynamics (MD) of the five model structures, and (iii) performed molecular docking studies of 15 ligands targeting different conformational rearrangements’ of DR subtypes. The binding energies, number of conformations as well as the distances between ligands and receptor interacting residues of the binding pocket were calculated for all complexes. The interaction between ligands and receptors were further analyzed using an in-house pipeline that takes advantage of BINding ANAlyzer (BINANA), a Python- implemented algorithm for analyzing ligand binding [39]. BINANA was shown to successfully atomically characterize key interactions between protein amino acids and ligand atoms, and as such it is a promising approach to map such interactions in GPCRs [39]. The main goal was to reveal new structural findings to help explain the mode of binding of the selected ligands and their selectivity for certain DR-subtypes.

## 2. Results 

### 2.1. Homology Models of Dopamine Receptors D_1_R-D_5_R Are Stable

The ligand-free D_2_-like homology models were generated using the resolved ligand-bound crystal structures of the D_2_R (PDBid: 6CM4) [36], D_3_R (PDBid: 3PBL) [37] and D_4_R (PDBid: 5WIU) [38] (over 90% identity). The 3D crystal structures of DRs are typically incomplete, lacking key regions for intracellular partner coupling such as intracellular loop 3 (ICL3). In contrast, D_1_-like DRs lack their own templates and therefore, the most suitable template to each DR was selected according to the percentage of similarity obtained upon sequence alignment by BLAST [40] in combination with ClustalOmega [41]. In fact, the D_3_R crystal structure was chosen as template for D_1_R (35.0% identity with BLAST and 39.5% with ClustalOmega), and the D_4_R crystal structure was chosen for D_5_R models (total similarity of 35.0% BLAST/39.1% ClustalOmega, check Materials and Methods section). We also calculated the similarity of the TMs in relation to the respective template and the results are summarized in Appendix A. All TMs of the D_2_-like subtypes showed almost 100% identity with their crystal structure templates, which is also in line with the total similarity. For D_1_R an average TM similarity with its template was 41.0%, compared to a total similarity of 39.5%, while for D_5_R 36.0% TM identity was calculated compared to the total similarity of 39.1%. For the D_1_-like subtypes no differences between the TM similarity and the total similarity with their template were obtained. Furthermore, for D_1_R the highest similarity between the model and its template was observed for TM1-3, whereas for D_5_R it was achieved for TM2, TM3 and TM7. Consequently, TM2 and TM3 seem to be very conserved among all DR subtypes. 

The combination of different metrics and scores were used to choose the most accurate models in order to perform MD and molecular docking: (i) the Discrete Optimized Protein Energy (DOPE) [42] score, MODELLER’s standard metrics, (ii) Protein Structure Analysis (ProSA-web) [43,44] and (iii) Protein Quality (ProQ)-LGscore and MaxSub score [45,46]. All final DR models (check Materials and Methods section) achieved LGscores > 4 and MaxSub scores > 0.5. The highest z-score was obtained for D_4_R model, whereas the lowest were counted for D_1_-like DR models. 

MD simulations of 100 ns were briefly run for each ligand-free modelled receptor and analyzed to confirm their stability. Root-Mean-Square-Deviations (RMSD) of Cα atom mean values ranged from 0.3 nm and 0.5 nm (Appendix A). Each DR model showed good overall stability. However, D_1_-like models showed slightly higher RMSD values than D_2_-like models: D_1_R (0.48 ± 0.07 nm) and D_5_R (0.49 ± 0.06 nm) vs D_2_R (0.35 ± 0.09 nm), D_3_R (0.34 ± 0.04 nm) and D_4_R (0.36 ± 0.09 nm). This behavior is justified by the higher homology scores attained for the D_2_-like subfamily. Additionally, RMSD of Cα carbons for individual TM was computed and their average values were listed in Appendix A. The low values obtained for TM’s RMSD further supports the good overall stability of the model structures.

### 2.2. Dopamine Receptor Binding Pocket Definition

In this work, we used the comprehensive review of Floresca and Schnetz [47], highly cited [48,49,50], as a base for the definition of the binding pocket of all DRs (Figure 2). Furthermore, by applying Ballesteros and Weinstein numbering (B&W) [51], the position of considered critical residues was more easily comparable between all receptors. This nomenclature is based on the presence of a highly conserved residue in each TMs [51], the so called X.50, in which X varies between 1 and 7 depending on the TM helix. The remaining residues were numbered according to their position relative to the most conserved one. 

Mutagenesis studies induced the believe that for dopamine binding, the endogenous agonist of the DR, a negatively charged aspartate (3.32Asp) forms an ionic bond interaction with the protonable amine of dopamine [2,50,52]. Moreover, it was shown that this effect was crucial for ligand binding and that this amino-acid was not only conserved among the DR, but also in all biogenic amine GPCRs [53,54]. Also, a serine microdomain in TM5 (5.42Ser, 5.43Ser, 5.46Ser) was considered as an important feature for dopaminergic binding in all DRs as it is believed that the serines form hydrogen bonds with the catechol hydroxyls of dopamine, increasing the binding affinity and orienting ligands in the orthosteric binding pocket [47,52,55,56,57]. While 5.42Ser seems to be critical, 5.43Ser plays a less important role [47]. A further microdomain, the aromatic microdomain, consisting of 6.48Trp, 6.51Phe, 6.52Phe and 6.55His/Asn was reported to trigger the activation of the dopamine receptor. All amino-acids in this microdomain share the same hydrophobic face in the water-assessable binding-site crevice, indicating that any reorientation of these residues by binding to a ligand would cause steric clashes and would therefore force the residues to reorient themselves in a domino-like fashion, which lastly leads to the so-called “rotamer toggle switch” [47,50,53,58]. In addition, 6.48Trp was reported together with 6.55His to stabilize the position of the ligand in the binding pocket via π-π-stacking [47,58]. Therefore, 6.48Trp and 6.55His as well as one phenylalanine (6.51Phe) were chosen for the docking protocol to mimic the ligand-binding on TM6. Dependent on the ligand properties other residues of TM3 were also considered, such as 3.33Val and 3.36Cys. 3.36Cys is believed to be part of a deeper subpocket below the orthosteric binding pocket (OBP) [36]. Additionally, Ericksen et al. reported that this cysteine was a relevant residue for benzamide binding [49]. Regarding 3.33Val, it was reported to show interaction with *N*-methylspiperdone by Moreira et al. [53] as well as with the methoxy ring of nemonapride, determined in the crystal structure of the D_4_R [38]. Different authors hypothesized that DRs have a secondary binding pocket (SBP) next to the OBP, which was confirmed by the resolved crystal structures together with computational analyses [37,38,59]. Crystal structures of D_2_R (PDBid: 6CM4) [36] and D_3_R (PDBid: 3PBL) [37] and computational data suggest that 7.43Tyr is also a crucial amino-acid for interaction in the SBP [17,36,37]. 2.57Val was shown to form a hydrophobic pocket for antagonists like clozapine and haloperidole [57]. However, since the OBP is widely explored through experimental, computational and crystal structure data, there could be other residues important in the SBP. Detailed information about the literature, mostly regarding D_2_-like DR can be found on Appendix A. In order to compare all DRs ligand-binding properties and specificity, we focused on the mentioned residues in the OBP. The residues considered flexible in the different dockings were listed in Methods and Materials section. 

### 2.3. Proof-of-Concept of Molecular Docking Success

Ten conformational rearrangements were chosen every 5 ns upon a 50 ns stabilization MD run. These 10 plus the initial model (time 0 ns) were then subjected to molecular docking of 15 different ligands. The results of the molecular docking were evaluated by AutoDock4.2, which ranks the possible binding positions by energy level and clusters these positions by RMSD of 2 Å. In addition, the total number of conformations (NoC) in these clusters were counted. Binding poses with more than five conformations per cluster were considered as a valid ligand position, despite the binding energy (BE) of this pose. All results of the docking can be checked in the Appendix A. 

As proof of concept, redocking of the co-crystalized ligands to the crystal structure templates of the D_2_R, D_3_R and D_4_R [36,37,38] was conducted (Appendix A). Receptors and ligands coordinates were retrieved from PDB files. Top clusters achieved a ligand pose equivalent to the pose in the correspondent crystal, presenting very small RMSD values. Lastly, these results were compared to the docking poses of the corresponding DR-models and ligands at time point 0 ns. The binding energies of the two sets were found to fall within a similar range. This is a further evidence of docking protocol reliability. 

For a general overview, dopamine docking was analyzed in detail (Figure 3A) as it is the endogenous ligand of the DRs and its binding mode is well-known compared to the other ligands [47]. However, we have to stress out the lack of a crystal structure with the dopamine-bound DR as the ligand’s structural properties are not suitable for crystallization (too small, not suitable for stabilizing a GPCR). We observed that the binding energy of D_2_R was the most stable at different analyzed MD conformations, while for the other subtypes it oscillated more frequently. Over time the average binding energy for all DRs was found to be at −9 kcal/mol. The highest NoC during all MD conformations were obtained for D_4_R and D_1_R (up to > 80 for D_4_R at 95 ns), whereas for D_2_R around 30 conformations were counted for all conformational arrangements (Figure 3B). Lastly, for all DRs complexed with dopamine, the first or the second cluster with the lowest binding energy also contained the highest NoC, indicating that the docking of dopamine was indeed stable and reliable (Appendix A). In summary, the binding energy and 3D positions of dopamine docking may demonstrate the binding mode of dopamine to DRs. According to Floresca and Schnetz, these features are crucial for dopamine’s binding affinity and DR activation but may not necessarily be true for all dopaminergic ligands (selective and non-selective) [47].

The binding position of dopamine to all DR complexes was stable over time namely, the protonable amine was always directed towards the aspartic acid on TM3 (3.32Asp) and the hydroxy groups were facing the serine microdomain (5.42Ser, 4.32Ser and 4.46Ser), in agreement with Floresca and Schetz [47] and Durdagi et al. [60] (Appendix A). As known from the literature dopamine’s interaction with the serine microdomain only typically requires two of the serines binding to the hydroxy groups [47]. At 0 ns dopamine was located planar in the OBP in the position described above. Notably, D_2_R and D_4_R hydroxyl groups were more directed towards serine microdomain (Appendix A). At 55 ns torsions were observed for dopamine bounded to all DR, which included a switch of interactions with the serines at TM5 for D_3_R, since it is known that dopamine is only capable of interacting with two of the three serines [47]. At 60 ns dopamine is shifted more to the serine and aromatic microdomain (TM6) for all DRs in a different manner. However, only at D_4_R a strong direction of dopamine’s protonable amine towards 3.32Asp was observed. At 65 ns dopamine bounded to all DRs was located again planar in the OBP (Appendix A). Small individual torsions were observed during the period of 70–90 ns (Appendix A). Interestingly, at 95 ns dopamine was strongly involved in the aromatic microdomain (TM6) at all DR, which was then vanished especially for D_3_R at 100 ns. The large decrease in D_4_R binding energy at 90 ns can be explained, by the approximation of dopamine to 3.32Asp distance from the serine microdomain (Appendix A). 

### 2.4. Docking of Various Ligands to DR Models

Since non-selective agonistic activity was already covered by dopamine docking, chlorpromazine was chosen as a non-selective antagonist [61,62]. Herein, we also selected the following ligands: SKF38393 as selective D_1_R agonist [27,30] and SCH23390 as D_1_-like DR antagonist [31,63], apomorphine as selective D_2_R agonist [60], 7-OH-DPAT as selective D_3_R agonist [23], nemonapride as D_2_R and D_3_R selective antagonist [64] and lastly haloperidole, due to its affinity for D_4_R [25]. This set of ligands was chosen as example of ligands with different DR selectivity (Table 1). The obtained binding energies and NoC in these clusters are summarized in Figure 4 (graphical output of the other ligands can be found in the appendix: Appendix A). 

For 7-OH-DPAT we observed a low and stable binding energy upon binding to all DRs. For apomorphine, a decrease in the binding energy was determined for D_2_R at 65 ns (−11 kcal/mol), whereas an increase at 85 ns was shown for D_4_R (−9 kcal/mol). Stable binding energies around −10 kcal/mol were observed for DR:nemonapride complexes, however a massive increase was observed for the D_5_R at 100 ns. For SCH23390, but not for SKF38393 the binding energy was stable over time at −9 kcal/mol for all DRs. The binding energy of SKF38393 at D_2_R and D_4_R increased at 85 ns. Haloperidole displayed the most interesting docking-profile: while the binding energies of DRs were stable at −10 kcal/mol, only for D_4_R a massive increase was observed at 55 ns and 80–90 ns into the positive range, meaning these binding positions were extremely unfavorable for haloperidole. Lastly, the chlorpromazine binding energy was increased only for D_1_R at 70 ns up to −3 kcal/mol. 

Similar to dopamine binding, the NoC of 7-OH-DPAT decreased at all DRs from 0 to 65 ns. For apomorphine, the lowest binding energies were obtained for D_1_R and D_2_R. Lesser NoC were counted for nemonapride in total at all DRs (max. 30 at 85 ns for D_2_R). The NoC for SKF38393 were the lowest over 70–85 ns period for D_1_R, D_2_R and D_3_R. In contrast to the BE of haloperidole, the NoC was found to be stable over time except for D_1_R with up to 40 conformations at 60 ns. In addition, most conformations were counted for the D_4_R especially at 0–70 ns for haloperidole. 

We also calculated the distance between the center of mass of the ligand and the alpha carbon of the binding pocket residues. Overall results of all ligand-residue measurements (Figure 5) showed that 3.32Asp was the closest residue for the majority of ligands. The ligand center of mass-residue alpha carbon distance was lower than 7–8 Å, particularly for D_1_R (<6 Å). We noted an increase in the distance between 3.32Asp and several ligands for D_4_R. The distances between 3.32Asp and both SKF38393 and SCH23390 ligands were larger at for D_3_R, D_4_R and D_5_R, but also D_2_R. This effect might occur due to the fact, that SCH23390 and SKF38393 are reported to be D_1_R-selective [30,63]. Subtype specific tendencies were observed for the serine microdomain. 5.42Ser was shown to be most distant at D_1_-like receptors and 5.43Ser for D_2_R and D_3_R (D_2_-like). These differences are less accentuated for dopamine, 7-OH-DPAT, apomorphine and bromocriptine. 

For 7-OH-DPAT, a known D_3_R selective agonist, distances between ligand and the defined pocket are higher for D_1_-like receptors and distinctive residue between D_2_-like seems to be 6.52Phe, that is closer to the ligand on D_3_R. The same pattern was visible with apomorphine, a selective D_2_R agonist, where distances in D_1_-like are higher, although distinction within D_2_-like family is less pronounced. Clozapine, sulpiride and risperidone are known as “dirty drugs” because of their non-selective profile, and for that reason none of these ligands showed distinctive differences between DR subtypes. Likewise, residues 3.32Asp and 3.33Val/Ile were the closest to clozapine in all five subtypes, suggesting that these residues are crucial for this ligand’s binding. Haloperidole, categorized as D_2_R selective antagonist with some activity on D_4_R, has distinctive differences between D_1_-like and D_2_-like family, being closer to the second (although within D_2_-like family there is no great differences on distances pattern). Spiperone and chlorpromazine have affinity for all DR subtypes, which agrees with the lack of significant differences in the measured distances. Finally, nemonapride and eticlopride, described as D_2_R/D_3_R selective antagonists, were located closest to D_2_-like DR residues compared to the D_1_-like DR, however it seemed as these two ligands demonstrated preference for D_4_R. 

### 2.5. The Type of Pairwise Interactions Between Receptor Amino-Acids and Ligand is Relevant for Binding

In-house scripts using the BINANA algorithm (software used in other non-GPCR studies [39,75,76,77,78]) were constructed to identify the type of interactions established between the ligands and binding pocket amino-acids [39]. We measured close contacts between receptor and ligands below or equal 2.5 Å and below or equal 4.0 Å, hydrogen bonds (HB), hydrophobic contacts (hydrocontacts) and salt-bridges (SB) as well as π-interactions, further subdivided into cation-π-interactions (cat-π), aromatic superpositions (π-π-stack) and perpendicular interactions of aromatic rings also referred to as edge-face-interactions (T-stack) [39]. For a first overview, all interactions despite their type and ligand were summarized and compared between the DR-subtypes (Results section at SI and Appendix A). Moreover, detailed mapping of pairwise interactions for each receptor is displayed in Figure 6. Appendix A show the change of interaction pattern over time for each ligand. Furthermore, the pairwise analysis highlighted the role of key receptor residues. By assorting those for each ligand at all DRs (time points summarized), patterns but also unique receptor-ligand interactions were highlighted (Appendix A).

### 2.6. 2.5 Å-Interactions

2.5 Å-interactions, very short (closer) contacts are especially relevant for ligand binding and are described in more detail herein. For dopamine the number of these interactions increased for D_1_-like DRs, while for 7-OH-DPAT the highest number of interactions observed in total only occurred for D_3_R. For bromocriptine, 2.5 Å-interactions were significantly higher for D_4_R. Also, haloperidole seemed to have a higher number of established interactions with D_4_R as well as eticlopride. Only risperidone had a higher number of interactions with D_2_R. Chlorpromazine had the lowest number of compared to all ligands with no preference for any DR-subtype. All in all, 2.5 Å-interactions seemed to be particularly relevant for the ligand binding to D_4_R.

### 2.7. Hydrogen Bonds and Hydrophobic Contacts 

Charge-reinforced hydrogen bonds are reported to be much stronger than the neutral hydrophobic contacts [79]. Moreover, it was reported that HBs determine the specificity of receptor-ligand binding [79]. Nevertheless, hydrophobic contacts also contribute to ligand binding, and a balance between HB and hydro contacts is required for drug-like molecules [79]. It was not surprising that a large number of hydro contacts was observed for all ligands, while HB were less common. Hydro contacts were preferably formed for D_1_R and achieved their lowest value for D_3_R. These contacts were particularly relevant for one ligand, bromocriptine (Figure 6). Moreover, a large hydrophobic network involving conserved and non-conserved residues of all TMs were found for all DRs (less pronounced for D_5_R). The “dirty drugs” were the second in line with the highest number of hydro contacts. 

Most interesting were the HB interactions. For dopamine a different set-up was presented at each DR. While the D_2_-like DRs and D_5_R HB were formed by the serine microdomain (5.42Ser, 5.43Ser and 5.46Ser), for D_1_R the serine microdomain was not involved at all. 3.32Asp appeared as interaction partner for all DRs. For D_5_R, an HB between 5.38Tyr and dopamine was stressed out as unique for all ligands. However, 5.38Tyr was found at the D_4_R to form HB with 7-OH-DPAT. Not more than 2 HB were found at any DR bounded to 7-OH-DPAT. Lastly, chlorpromazine does not seem to form any HB in any DR complex. 

### 2.8. Salt-Bridges

Most stable SB interactions were unsurprisingly achieved by dopamine for all DR sub-types. 7-OH-DPAT, SB were found for D_1_R (three in total), while for the other subtypes, contacts ranged between one and three over time. The same trend was observed for nemonapride and SKF38393. SCH23390 formed the highest number of SB with D_5_R and with D_2_R between 70 and 85 ns. Haloperidole seemed to establish a higher number of SB with D_1_-like DR and D_2_R, while none were formed with D_3_R and D_4_R. Spiperone seemed to preferably form SB with D_1_-like DRs. The following ligands did not form any salt-bridges at any time point: apomorphine, bromocriptine, clozapine, risperidone, aripiprazole and chlorpromazine. 

Undoubtedly, 3.32Asp was always involved in the establishment of SB in all DRs. However, at D_1_R, 74Pro located on ECL1 appeared also to establish relevant SB interactions. In addition, D_3_R SB-bonding for spiperone was found to occur involving 1.44Leu and 75Ser (ECL1) rather than 3.32Asp. All in all, salt-bridges were found to be highly conserved regarding the residues involved. 

### 2.9. Cat-π- and π-π-Stacking Interactions 

Cationic-π and π-π-stacking are considered as natural key non-covalent interactions [80]. They are important as solitary effects, but also their interplay omnipresent in many biological systems [81]. In the DR-ligand system frequent oscillations between different receptor conformations were noted for some ligands, depicted in Appendix A. 

Dopamine, for example, showed the highest cat-π-interactions for D_2_R, oscillating from 2–4 interactions/time point. Cation-π-interactions seemed to be more relevant for D_4_R and were less common and mainly formed by conserved residues on TM6 (6.42Gly, 6.31Thr, 6.30Glu, 6.39Val) for D_1_R. Bromocriptine (3.28Trp, 6.51Phe), nemonapride (6.48Trp, 6.51Phe, 6.52Phe), sulpiride (2.61Lys, 6.48Trp, 6.51Phe) and SCH23390 (6.48Trp) showed one cat-π-contacts to D_5_R each. For risperidone, cat-π-interactions were mainly formed with D_4_R, while π-π-stacking was mostly related to D_3_R complexes. Aripiprazole seemed to preferably form cat-π-interactions with D_4_R, while increasing π-π-stacking-interactions were observed with D_1_R between 65 and 80 ns. Haloperidole seemed to prefer π-π-stacking-interactions with D_2_R, maybe important for its selectivity towards this receptor. For chlorpromazine, no cat-π-interactions were observed at D_1_-like DRs (D_1_R and D_5_R), while many interactions were counted with D_2_R between 65 and 75 ns, with D_3_R at 95 ns and with D_4_R at 60 ns. 

The π-π-interactions were rather rare compared to the other interaction types. Some ligands did not form π-π-stacking interactions with DR subtypes (e.g., D_1_R binding to dopamine, 7-OH-PAT and sulpiride; D_2_R binding to sulpiride either, D_4_R binding to eticlopride and haloperidole; D_5_R binding to nemonapride). It was also obvious that the residues of the aromatic microdomain (6.48Trp, 6.51Phe, 6.52Phe, 6.55His) were responsible for the majority of ligands interactions to all DRs. However, different residue partners were determined for π-π- compared to T-stacking such as residues from TM5 (5.38Tyr, 5.47Phe). For aripiprazole, residues 7.43Tyr (D_2_R-D_4_R) and 7.34Thr (D_1_R) seemed also to be important for this type of interaction. Most interesting was the interaction pattern for sulpiride: while for D_1_R and D_2_R no π-π-stacking was detected, for D_3_R and D_5_R only a few residues seemed to be relevant (2.43Val, 2.44Val, 2.48Val, 38Thr, 5.38Phe, 6.51Phe, 6.52Phe for D_3_R; 3.28Trp and 6.48Trp for D_5_R) while for D_4_R, 27 residues from all TMs were involved in contact network formation. This may be explained by the different possible binding poses of sulpiride on the different D_4_R conformations. 

### 2.10. T-Stacking Interactions

T-stacking-interactions were similar to cat-π- and π-π-interactions, yet more frequent fluctuations in the number of interacts between ligands and receptor were observed in total. Especially for risperidone, which showed the highest number of T-stacking-contacts, preferably with D_2_R. Haloperidole and spiperone also seemed to have a D_2_R-preference, while chlorpromazine formed a large number of interactions with D_5_R. Despite the ligand, T-stack-contacts involved mainly conserved residues (6.39Val, 6.42Gly, 6.43Val) or residues from the aromatic microdomain (6.48Trp, 6.51Phe, 6.52Phe, 6.55His). An exception was bromocriptine and sulpiride for D_2_R, haloperidole for D_4_R and spiperone for D_5_R. Unique interactions were found for risperidone binding to D_4_R with 6.44Phe and for chlorpromazine binding to D_1_R with 6.30Glu. However, other residues from other TMs were also involved in forming T-stack-contacts: for example, 7-OH-DPAT unique interaction with 2.47Ala and SKF38393 with 35Ala (ICL1) were found at D_3_R. For risperidone, a unique interaction with 231Phe (ICL3) was determined for D_1_R. Whereas for spiperone 1.35Tyr and 159Ile (ECL2) seemed to be relevant for D_4_R, 2.14Tyr was relevant for chlorpromazine coupling. 

However, TM7 residues were also involved in T-stack-formation: 7.34Thr (D_1_R) and 7.35Tyr (D_2_-like)/7.35Phe(D_5_R), 7.43Tyr(D_2_-like). Residues on TM2 were also relevant for T-stack-formation (2.41Tyr, 2.43Val, 2.45Ser, 2.46Leu, 2.47Ala, 2.50Asp) but only for D_3_R. For D_4_R and D_5_R, only residues from TM6 and TM7 were involved in T-stack-contacts, except for SKF38393 where 5.47Phe was relevant for binding to D_4_R. Lastly, for D_1_R and D_2_R TM3 (3.28Trp(D_1_R)/3.28Phe(D_2_R)) residues also established meaningful interactions with nemonapride, sulpiride, SCH23390, aripiprazole and spiperone. Although these residues (especially on TM2 and TM7) are more related to the SBP than to the OBP (herein TM6 is the most relevant TM), contact formation was also observed for smaller ligands (7OH-DPAT, SCH23390, SKF38393). It was not expected that these ligands would access the SBP. Noteworthy is also the fact, that dopamine exclusively formed T-stack-contacts with the conserved aromatic microdomain for all DR. Finally, it was also obvious that the variety of T-stack-contacts was also limited by the number of aromatic rings of the ligand (e.g., dopamine only contacted 3 different sequential residues). In brief, our results also pinpoint for the fact that T-stacking-interactions seem to be relevant for large ligands, primary in antagonists binding than in agonists case.

## 3. Discussion

One of the major research efforts in the research of dopamine receptors is the design of DR-subtype selective ligands [82]. However, most predictive studies have been performed on D_2_R ligand specificity, as this receptor is the most crucial in neurotransmission [17,57,83]. Herein, we present a comprehensive in silico approach, which reveals important interactions between DRs key residues and ligands in a more detailed way when compared with available literature [55,57,59,60,84,85].

### 3.1. Validation of the In Silico Pipeline

Homology modeling and TM definition of all DR subtypes showed that there were smaller structural differences among the “classical” TMs (TM3, TM5, TM6), which are known to be key for ligand binding. Yet, as expected, structural differences between the subtypes were observed in the intracellular and extracellular loops, where some are important for ligand binding (ECL2) or for intracellular signaling (ICL2 and ICL3) [86]. This was particularly true for D_1_-like receptors, due to their much larger ICL3. Although no crystal structure was available for the D_1_-like DRs, the high sequence similarity among all DR helped to find suitable models for molecular docking. Validating the docking performance by low binding energies and high NoC by cluster also showed that the homology modeling-docking approach was suitable and reproducible. In fact, the combination of the different software and in-house scripting resulted in a straightforward in silico approach which can certainly be applied for studying other GPCRs. Data is also in line with experimental information, which corroborates the conceptual framework of this analysis protocol [47].

### 3.2. Pairwise Interactions Analysis Was Able to Determine Key Amino-Acids and Types of Interaction

A clear D_2_-like selectivity or binding preference was only found for apomorphine, while for others either D_2_R and D_5_R seemed to form a lower number of 4 Å-interactions such as nemonapride (D_2_R/D_3_R-antagonist [87]), SCH23390 (D_1_-like antagonist [88]), SKF38393 (D_1_R-antagonist [30]) or D_1_R and D_4_R were highly preferred (higher number of meaningful interactions). In other cases, such as for eticlopride (D_2_R/D_3_R antagonist [37]) and spiperone (D_2_R-antagonist [64]), the D_3_R was the least attractive DR for interaction. It was shown that the “classical” conserved residues e.g., 3.32Asp, the serine microdomain 5.42Ser, 5.45Ser, 5.46Ser and the aromatic microdomain 6.48Trp, 6.51Phe, 6.52Phe, 6.55His were relevant for all ligands and formed specific interactions, electrostatic (cat-π, π-π, T-stack), salt-bridges and hydrogen bonds. These residues were omnipresent in all our analyses. Yet, the distances for the most conserved OBP residues (3.32Asp, serine residues and 6.48Trp), distinct differences were observed between agonists and antagonists. For example, dopamine was constantly close to OBP, indicating its receptor activating properties as described by Floresca and Schnetz [47], while risperidone was found distant from these residues according to its antagonistic properties. This was also the case for the other antagonists such as haloperidole, nemonapride and the biased ligand aripiprazole. In addition, we described other TM residues involved in binding of these ligands, as previously described by Kalani et al. for D_2_R [57]. 

It was not surprising that the “classical” TMs, e.g., TM3, TM5, TM6 and TM7 were involved in many different interaction types. TM3 residues such as 3.35Cys, 3.36Ser, 3.33Val or 3.33Ile and 3.39Ser were often found forming different interactions with different ligands. This was also in concordance with previous studies regarding the involvement of other conserved residues on TM2 and TM7 (and TM3) [57,82,83,85], which was also described as part of a SBP only assessable for ligands with piperazine-moieties [59]. Residues on TM4 were not contributing to receptor-ligand interaction, except for D_4_R complexes. By comparing large ligands such as spiperone or haloperidole with rather compact ligands such as dopamine, SCH23390 or clozapine, it was possible to point out a larger number of TM1 and TM2 residues involved in establishing meaningful interactions. Author’s had already hypothesized that these residues could belong to a SBP, only accessed by large ligands [57,85]. Furthermore, there was a clear higher network contact formation with D_4_R. Except for that fact that the D_4_R is physiologically distant compared to D_2_R and D_3_R [82], no further explanation could be found for this trend.

A systematic study by De Freitas and Schapira [89] showed that the most frequent type of non-covalent interactions for protein-ligand complexes were hydrophobic contacts, followed by hydrogen bonding, π-stacking, salt-bridges, amide-stacking (corresponds to T-stack) and lastly cation-π-stacking. The same ranking of frequency of interaction type was found in our study. As also described by Davis and Teague [79] hydrophobic contacts are the most common type of receptor-ligand-interactions as they not only enhance binding affinity but also are sometimes favored over tight, charged hydrogen bonds [79]. In addition, they can be formed with different ligand-atoms such as carbons, halogens or sulphurs [89]. As reviewed in Davis and Teague [79] most docking studies fail to count in the hydrophobicity for their ligands. However, the balance between polarity (causing hydrogen bonds) and lipophicity (causing hydrophobic contacts) is the main drive to make a ligand “drug-like” [79]. Our study was successful to determine not only the hydrogen bonds but also the large hydrophobic network of each “drug-like” ligand (as well as of the marketed drugs). Hydrophobic contacts appeared to form a huge network of conserved and non-conserved residues that stabilized ligand positions during binding. This network was spread across TM2-TM3-TM7. Residues from TM1 and TM2 were shown to be relevant for binding large ligands such as nemonapride. Lastly, T-stacking interactions revealed as especially relevant for some large ligands such as apomorphine, risperidone or aripiprazole. 

Conserved residues in the OBP were found to be clustered in microdomains, stabilizing ligand-binding through the formation of a HB network. Indeed, HBs where mostly mediated by the serine microdomain (5.42Ser, 5.34Ser and 5.46Ser especially at D_2_R and D_5_R). Interestingly, these residues were not relevant for D_1_R, although a study by Hugo et al. mentioned 5.46Ser as key residue for activating D_1_R [90]. In this study, 3.37Trp was also proposed to be mediator of the D_1_R-activation [90]. We were not able to confirm these findings in our study, only bromocriptine and spiperone were interacting 3.37Trp at D_1_R, while at D_5_R we did not observe any interaction with this residue. 3.37Thr D_2_R was found to interact with 7-OH-DPAT, indicating that these residues may not be D_1_R-specific. Salt-bridges were exclusively formed by 3.32Asp but appeared to involve also residues from ECL1 for spiperdone for D_1_R and D_3_R. For “bulky” ligands such as clozapine or bromocriptine no salt-bridges were formed. 

Frontera et al. reported that the strength of cation-π-interactions is also influenced by the presence of weaker interactions such as hydrogen or hydrophobic bonds [81]. For instance, it is well known that H-bonding is highly contributing to the bond strength of π-stacking [81]. But not only weaker interactions benefit π-interactions, cat-π and π-π-stacking were also found to be cooperative for each other [81]. Such combinations where cat-π and π-π-stacking were simultaneously present, were indisputably found for D_2_-like rather than for D_1_-like DRs. In addition, these residues and those of the TM6 aromatic microdomain (6.48Trp, 6.51Phe, 6.52Phe, 6.55His/Asn) were mostly involved in forming π-interactions (cat-π, π-π or T-stack). Phenylalanine, tyrosine and tryptophan interacting ligands could indeed be further extended in order to design a new selective SAR for D_5_R, as they were found to be exclusively involved in π-interactions and π-stacking formation at this DR subtype. Since for the D_1_R-like DR SCH23390 and SKF38393 are the only known selective ligands, a closer look at the interacting residues of these ligands revealed that cat-π-interactions (6.30Glu, 6.39Val, 6.42Gly) were only present at the D_1_R for SCH23390, the antagonist at the D_1_-like DR [88]. Moreover, these residues were not the “classical” TM6 residues usually involved in binding, while this was true for the other ligands. This encourages the search for D_1_R- or D_5_R-selective ligands, which should ideally form cat-π-interactions with certain amino-acids, as they were found in this ligand set. From a structural basis SCH23390 and SKF38393 are more related to the benzodiazepines, compared to the other ligands that are either small molecules or longer ligands with piperidine moieties [91]. Lastly, another difference found between SCH23390 and SKF38393 binding to D_5_R were that SKF38393 established more interactions with residues from different TMs and a variety of neighboring residues of the “classical” interacting residues; whereas SCH23390-receptor-interactions were more limited to a smaller number of residues. These observations were not found for both ligands at the D_1_R. Reported by Bourne, who discovered SCH23390, this compound is the 3-methyl, 7-chloro analogue of the D_1_R agonist SKF38393, which is furthermore enantioselective [88]. In addition, it was stated that the phenyl ring in the benzodiazepine-derivatives and the receptors was involved in electrostatic forces, important for binding [88,92]. Mapping the full electrostatic potential of the D_5_R using ligands with benzodiazepine properties may be useful to find D_5_R-selective SAR.

In order to find future SARs for DRs and improve subtype selectivity, we should not only considerer the known “classical” residues and binding motifs such as the “DRY” motif, but also conserved neighboring amino acids as shown herein. For sure, this would improve the treatment with antipsychotics of many patients. 

## 4. Materials and Methods 

### 4.1. Homology Modeling

#### 4.1.1. General Approach 

The apo-DR models were generated with MODELLER 9.19 (version MODELLER 9.19, released Jul 25th, 2017) (University of California San Francisco, San Francisco, CA, USA) [93]. For D_2_-like receptors we used their corresponding crystal structures as templates: the D_2_R complexed with risperidone (PDBid: 6CM4) [36], the D_3_R complexed with D_2_R-antagonist eticlopride, (PDBid: 3PBL) [37] or D_4_R complexed with D2R/D3R-antagonist nemonapride (PDBid: 5WIU) [38]. Depending on the sequence similarity obtained with Basic Local Alignment Search Tool (BLAST, NCBI, Rockville, MD, USA) [40] and ClustalOmega (EMBL-EBI, Cambridgeshire, UK) [41] and listed in Table 2, either D_3_R (for the D_1_R) or D_4_R (for the D_5_R) were chosen as template to model the D1-like DRs. Due to the length of the ICL3, this was cut and substituted with two or four alanine residues, for D2- and D1-like receptors. Water and co-crystalized compounds were removed from the template structures. In the modeling protocol the lengths of the TMs and the perimembrane intracellular helix (HX8) were specified. In addition, disulphide bonds were constricted in the known pairs of cysteines, in particular between 3.25Cys and a non-conserved cysteine in ECL2 and between two non-conserved cysteines in the ECL3. Furthermore, loop refinement was performed for extracellular and intracellular loops for all DR using the module “loop refinement” of MODELLER 9.19. The number of models calculated with MODELLER [93] was set to 100.

#### 4.1.2. Model Evaluation/Methods of Quality

There are several approaches to validate homology models such as built-in metrics of open-source [52] and licensed softwares [94]. In a preliminary study we experienced [50] that the combination of different independent metrics provided adequate models suitable for molecular docking. For instance, the combination of MODELLER’s metrics [95], ProSA-web [43,44] and ProQ [45,46] revealed to be a promising and reliable protocol to create valid models for molecular docking.

Discrete Optimized Protein Energy (DOPE) [42] scores are MODELLER’s standard metrics and were utilized in combination with visual inspection to initially remove incorrect models. DOPE is specific for a given target sequence, e.g., it accounts for the finite and spherical shape of native protein states with the lowest free energy [42]. It should be noted, that although DOPE is not an absolute measure, it helps to rank the proposed models. Then, out of a small set of potential candidates (selection of 5–10), ProSA web service [44] and the online ProQ prediction server [46] were used to determine the final models with the best combination of scores. For the z-score provided by ProSA-web analysis values around −4 are suggested as acceptable. It was only used for error recognition, as it indicates overall model quality with respect to an energy distribution derived from random conformations for globular proteins [44] The ProQ analysis (LGscore [95] and MaxSub [96]) provides absolute measures based on a neural network, which were set as base for the more detailed evaluation of the models. Regarding the LGscore, values > 3, for MaxSub values > 0.5 are typically considered as “good”. Additionally, ProQ allows to include secondary structure information calculated with PSIPRED [97], improving the prediction accuracy and increasing the model quality up to 15%. The ProQ analysis was only carried out, if z-scores around 2–4 were achieved using the ProQ protocol. 

We could not compare our models with other authors as metrics scores are mostly not shown [43,98]. D_1_-like models, without a known crystal structure and D_2_-like models for which there are 3D crystal structure, showed similar quality (Table 3). 

### 4.2. Molecular Dynamics

#### 4.2.1. System Setup

It is well known that GPCRs take an infinite number of conformations over time. As such we performed MD simulations of modelled apo-forms to verify the effect of punctual fluctuations into the overall binding arrangements of ligands. Before setting up the system, the selected DR models were subjected to the Orientations of Proteins in Membranes (OPM) web-server [99,100,101,102] to calculate spatial orientations respecting to the Membrane Normal defined by the z-axis. In addition, the state of titratable residues was calculated by Propka 3.1 [103,104] within the PDB2PQR web-server [105] at a pH of 7.0. The prepared receptor structures were inserted into a previously constructed lipid bilayer of POPC: Cholesterol (9:1). Insertion of the receptors in the membrane was performed with *g_membed* package of GROMACS [106]. System was then solvated with explicitly represented water. Sodium and chloride ions were added to neutralize the system until it reached a total concentration of 0.15 M. The final systems dimensions were 114 × 114 × 107 Å and included approximately 370 POPC, 40 cholesterols, 125 sodium ions, 139 chloride ions and 28500 water molecules, with small variations from receptor to receptor.

#### 4.2.2. Molecular Dynamics Simulation Protocol 

CHARMM36 force field, with an implemented CMAP correction, was used for ions, water (TIP3P model), lipids and protein parameters [107]. Prior to MD simulation, the systems were relaxed to remove any possible steric clashes by a set of 50,000 steps of Steepest Descent energy minimization. Equilibration was performed afterwards as following: the system was heated using Nosé-Hoover thermostat from 0 to 310.15 K in the NVT ensemble over 100 ps with harmonic restraints of 10.0 kcal/mol. Then systems were subjected through a first step of NPT ensemble of 1 ns with semi isotropic pressure coupling and a pressure of one bar. Further equilibration was performed with sequential release of membrane lipids and protein’s atoms with a final step of NPT ensemble with harmonic restraints on the protein of 1.0 kcal/mol, for a total of 5 ns of restrained equilibration.

MD simulations of all DR models were performed with the periodic boundary condition to produce isothermical-isobaric ensembles using GROMACS 5.1.1 [106]. The Particle Mesh Ewald (PME) method [108] was used to calculate the full electrostatic energy of a unit cell in a macroscopic lattice of repeating images. Temperature was regulated using the Nosé-Hoover thermostat at 310.15 K. Pressure was regulated using the Parrinello-Rahman algorithm. The equations of motion were integrated using leapfrog algorithm with a time step of 2 fs. All bonds, involving hydrogen atoms within protein and lipid molecules were constrained using the LINear Constraint Solver (LINCS) algorithm [109]. Additionally, a cut-off distance of 12 Å was attributed for Coulombic and van der Waals interactions. Then a single independent simulation of 100 ns was initialized from the final snapshot of the restrained equilibration from each DR, for a total of 5 simulations. Trajectory analysis was performed by in-house scripting using GROMACS [106] and Visual Molecular Dynamics (VMD) [110]. Trajectory snapshots were saved every 5 ns. The snapshots after the first 50 ns MD stabilization were used for molecular docking studies.

### 4.3. Molecular Docking

#### 4.3.1. Ligand Dataset

The following ligands were docked into the receptor decoys: dopamine, 7-hydroxy-*N*,*N*-dipropyl-2-aminotetralin (7-OH-DPAT), apomorphine, bromocriptine, clozapine, nemonapride, sulpiride, SCH23390, SKF38393, eticlopride, risperidone, aripiprazole, haloperidole, spiperone and chlorpromazine (Table 1). All structures were obtained from the DrugBank database (https://www.drugbank.ca) or from ChemSpider (http://www.chemspider.com) [111]. 

#### 4.3.2. Docking Procedure

DR binding pocket was defined in several experimental and computational studies [2,47,52,55,57,59,85]. Herein, we used the comprehensive review by Floresca and Schetz [47] as a base for exploration of the DR binding pocket, since it contains detailed experimental data. A summary of the procedure can be better reviewed in Bueschbell et al. [50]. AutoDock4.2 (version AutoDock 4.2.6, released in 2009) was used to perform ligand docking [112]. DR hydrogens were added and Kollman united atom charges were assigned. Hydrogens were also added to the ligand and Gasteiger-Marsili was used to calculate charges. Before docking an energy, grid was created using AutoGrid (version AutoGrid 4.2.6, released 2009) with a box-size varying with the times step and ligand. For each docking simulation 100 independent Lamarckian genetic algorithm (LGA) runs were performed with the number of energy evaluations set to 10.000.000, the population size set to 200 and the maximum number of generations set to 27.000. Default settings were maintained for the rest of the parameters. Docked conformations within a RMSD of 2 Å were clustered. The most populated and lowest energy cluster (Gibbs free energy of binding) was used for conformational analysis. To find the local energy minimum of the binding site with a limited search space to that region, a low-frequency local search method was used. The 100 conformations obtained from docking were clustered by low-energy and RMSD. The top-ranked conformations within the best 3 clusters were visually inspected. The docking parameters were not changed for any ligand, only the residues treated as flexible in the docking protocol differed between the ligands. The flexible residues for each DR model are summarized in Table 4.

#### 4.3.3. Analysis of Molecular Docking

In this study, 15 DR ligands were docked to the homology model and to different conformational arrangements retrieved at every 5 ns for the 55–100 ns range for each DR simulation (825 dockings in total). All distances between the center of mass of the ligand and the alpha-C-atom (Cα) of the residues, treated as flexible in the docking protocol, were calculated using in-house PyMOL scripts [2,17,47,52,57,59] as well as previously published work [50]. We also develop in-house BINANA scripts to predict the main receptor-ligand interactions [39]. BINANA is an open-source python-implemented algorithm which uses output files from AutoDock [112] for the analysis of interactions and visualizes them in the free molecular-visualization program VMD [110]. Key binding characteristics such as hydrogen bonds, hydrophobic contacts, salt-bridges and π-interactions were calculated with BINANA. 

## Figures and Tables

**Figure 1 molecules-24-01196-f001:**
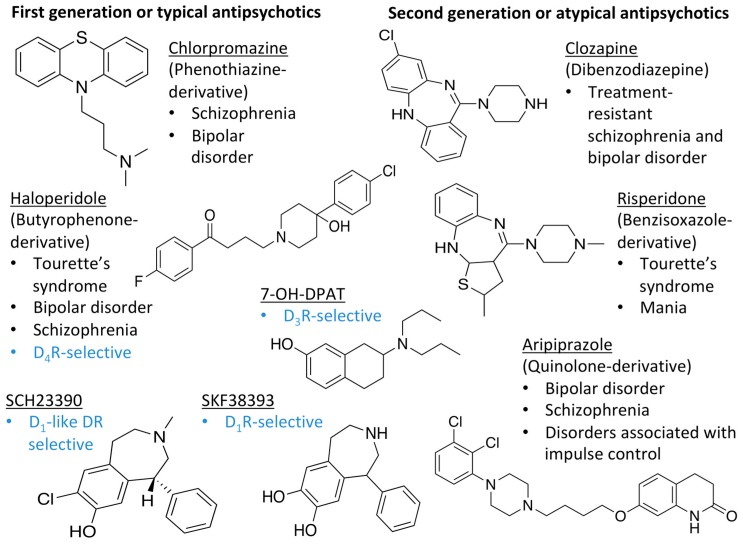
Examples for first-line treatments of neurological diseases and selective dopamine receptor ligands. Drugs are classified in typical and atypical antipsychotics [19]. The targets of the selective ligands haloperidole, 7-OH-DPAT, SCH23390 and SKF38393 were colored in blue.

**Figure 2 molecules-24-01196-f002:**
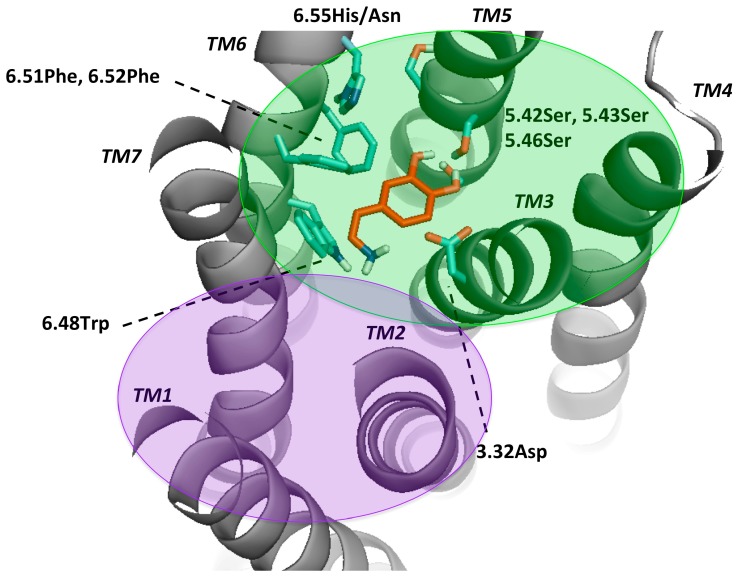
Key residues for molecular docking of Dopamine to DR models. Residues were designated in bold and colored in cyan, TMs in bold and italic. This set-up was kept for Appendix A and highly corresponds to the definition of the binding pocket of Floresca and Schnetz [47]. The area of the orthosteric and secondary binding pockets were colored and violet, respectively.

**Figure 3 molecules-24-01196-f003:**
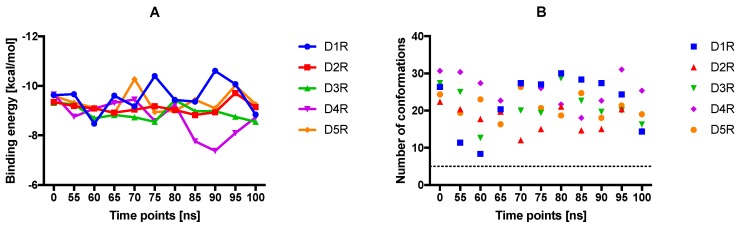
Results of the molecular docking of dopamine to all DR subtypes at all MD time steps. (**A**) The average binding energy of the three lowest energies of dopamine was calculated. (**B**) The mean of the number of conformations of the three clusters with the lowest binding energies are shown for each time point and receptor.

**Figure 4 molecules-24-01196-f004:**
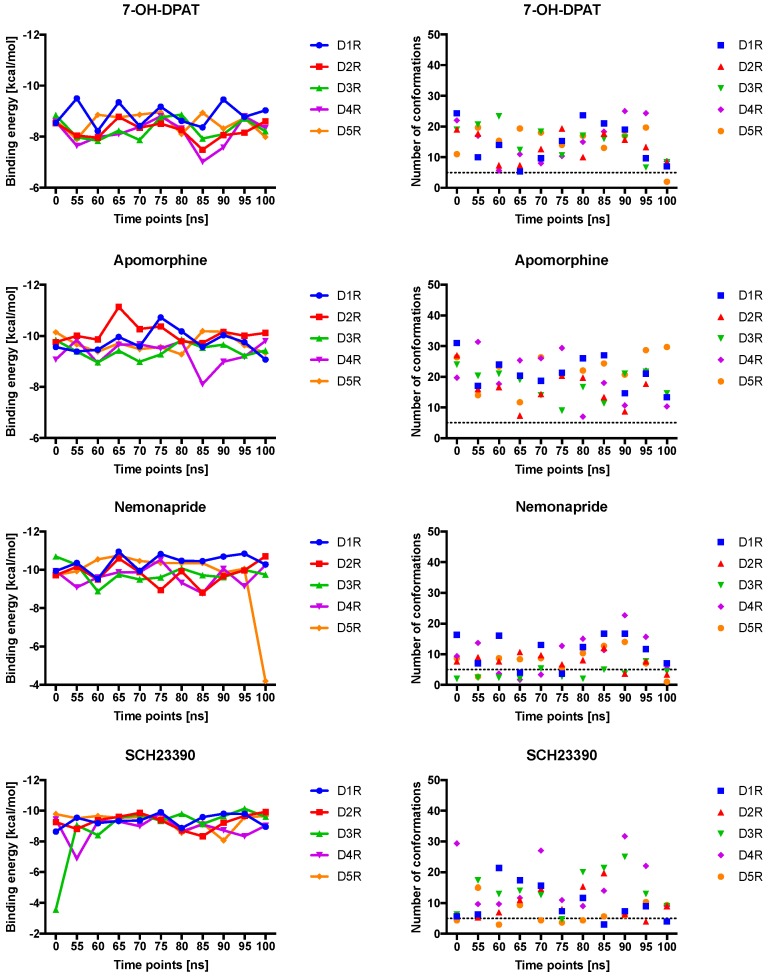
Results of the molecular docking of 7-OH-DPAT, apomorphine, nemonapride, SCH23390, SKF38393, haloperidole and chlorpromazine for all DR subtypes at time points [ns]. The average of the three lowest binding energies of dopamine were calculated in the left plots. The mean of the number of conformations of the three clusters with the lowest binding energies were plotted for each time point and receptor (right plot).

**Figure 5 molecules-24-01196-f005:**
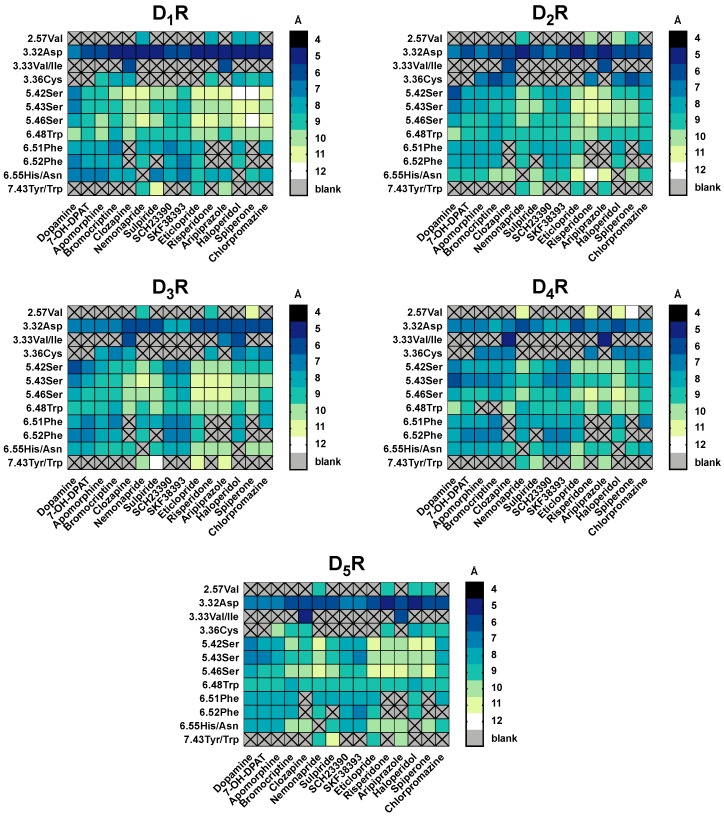
Summary of the distances between ligands and residues used in molecular docking for all DR subtypes. For each ligand-residue-distance [Å], we calculated the mean of all time points of the conformational models (11) of the three best docked clusters ranked by binding energy [kcal/mol] Noteworthy is that not all ligands were set to interact with all residues shown in the x-axis in the molecular docking. (e.g., only clozapine and aripiprazole were set to interact with 3.33Val). The distances are color coded: while dark colors indicate short distances, light colors indicate wider distances.

**Figure 6 molecules-24-01196-f006:**
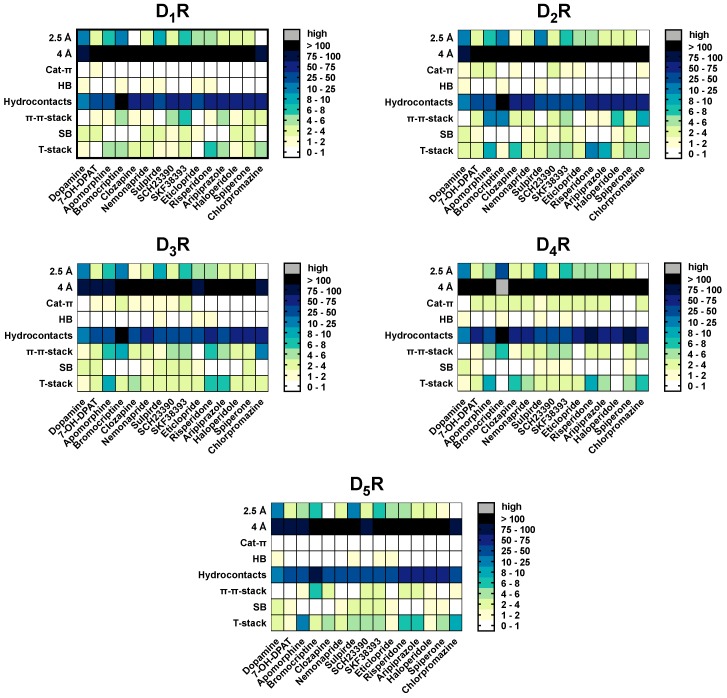
Interaction types counted for each ligand at DR-subtypes. Data are summarized for each ligand at all time points. Total numbers of the contacts for each interaction type are color-coded: few interactions were colored white, while many interactions were colored dark. Grey cells indicate that these values are outside the scale, which was only the case for bromocriptine at the D_4_R with 360 four Å-interactions.

**Table 1 molecules-24-01196-t001:** Ligands used for molecular docking and information on their function.

LIGAND		FUNCTION	BP	REFERENCES
**DOPAMINE**	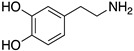	Endogenous agonist of all DR	OBP	[47,52,65]
**7-OH-DPAT**	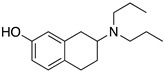	Synthetic D_3_R selective agonist	OBP	[47,65,66]
**APOMORPHINE**	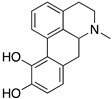	D_2_R selective agonist	OBP	[47,52,65]
**BROMOCRIPTINE**	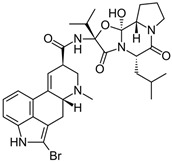	D_2_R selective agonist	OBP	[47,65]
**CLOZAPINE**	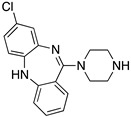	“Dirty drug”, multiple receptor binding	OBP	[47,65,67,68]
**NEMONAPRIDE**	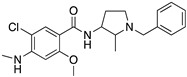	D_2_R/D_3_R selective antagonist	OBP + SBP	[38,47,55,65]
**SULPIRIDE**	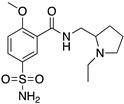	“Dirty drug”, multiple receptor binding	OBP + SBP	[47,65,66]
**SCH23390**	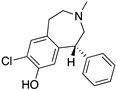	D_1_R antagonist	OBP	[31,47,65,69]
**SKF38393**	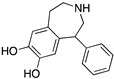	D_1_R selective agonist	OBP	[31,47,65,70]
**ETICLOPRIDE**	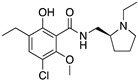	D_2_R/D_3_R selective antagonist	OBP + SBP	[37,66]
**RISPERIDONE**	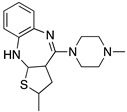	“Dirty drug”, multiple receptor binding	OBP+SBP	[36,47]
**ARIPIPRAZOLE**	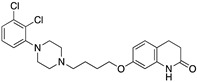	Partial D_2_R agonist, D_2_R/D_3_R heterodimer antagonist	OBP + SBP	[66,71]
**HALOPERIDOLE**	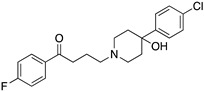	D_2_R selective antagonist, D_4_R antagonist	OBP+SBP	[47,65,67,72,73]
**SPIPERONE**	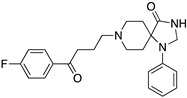	Affinity for all DR	OBP + SBP	[47,65,66]
**CHLORPROMAZINE**	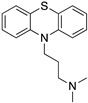	Antagonist on all DR	OBP	[47,65,74]

Abbreviations: DR-dopamine receptors, BP-binding pocket, OBP-orthosteric binding pocket, SBP-secondary binding pocket.

**Table 2 molecules-24-01196-t002:** Identity between DRs in study and their corresponding templates calculated with BLAST [40] and ClustalOmega [41].

DOPAMINE RECEPTOR	TEMPLATE	BLAST [%]	CLUSTALOMEGA [%]
**D_1_R**	3PBL	35.0	39.5
**D_2_R**	6CM4	97.0	100.0
**D_3_R**	3PBL	93.0	99.3
**D_4_R**	5WIU	93.0	100.0
**D_5_R**	5WIU	35.0	39.1

**Table 3 molecules-24-01196-t003:** Metrics and scores of the final DR homology models used herein.

DR	DOPE	LGscore	LGscore + PSIPRED	MaxSub	MaxSub + PSIPRED	z-Score
D_1_R	−39070.82	2.53	4.26	0.18	0.53	−2.14
D_2_R	−39284.66	2.52	4.22	0.21	0.52	−2.22
D_3_R	−39458.37	3.14	4.19	0.27	0.55	−3.12
D_4_R	−36738.05	3.33	4.25	0.25	0.59	−3.90
D_5_R	−38356.05	2.60	4.14	0.15	0.57	−1.49

**Table 4 molecules-24-01196-t004:** Flexible residues used in the molecular docking different ligands.

LIGAND	FLEXIBLE RESIDUES IN B&W NUMBERING
**DOPAMINE**	3.32Asp, 5.42Ser, 5.43Ser, 5.46Ser, 6.48Trp, 6.51Phe, 6.52Phe, 6.55His/Asn
**7-OH-DPAT**	
**APOMORPHINE**	3.32Asp, 3.36/3.35Cys, 5.42Ser, 5.43Ser, 5.46Ser, 6.48Trp, 6.51Phe, 6.52Phe, 6.55His/Asn
**BROMOCRIPTINE**	
**CLOZAPINE**	3.32Asp, 3.33Val, 3.36Cys, 5.42Ser, 5.43Ser, 5.46Ser, 6.48Trp, 6.55His/Asn
**NEMONAPRIDE**	2.57Val, 3.32Asp, 5.42Ser, 5.43Ser, 5.46Ser, 6.48Trp, 6.51Phe, 6.52Phe, 7.43Tyr
**SULPIRIDE**	3.32Asp, 6.48Trp, 5.42Ser, 5.43Ser, 5.46Ser, 6.55His/Asn, 7.43Tyr, 6.51Phe
**SCH23390**	3.32Asp, 6.48Trp, 5.42Ser, 5.43Ser, 5.46Ser, 6.55His/Asn, 6.51Phe, 6.52Phe
**SKF38393**	
**ETICLOPRIDE**	3.32Asp, 6.48Trp, 5.42Ser, 5.43Ser, 5.46Ser, 6.55His/Asn, 7.43Tyr, 6.51Phe, 6.52Phe
**RISPERIDONE**	3.32Asp, 6.48Trp, 3.36Cys, 6.55His/Asn, 2.57Val, 5.42Ser, 5.43Ser, 5.46Ser
**ARIPIPRAZOLE**	3.32Asp, 6.48Trp, 3.33Val, 5.42Ser, 5.43Ser, 5.46Ser, 7.43Tyr, 6.55His/Asn
**HALOPERIDOLE**	3.32Asp, 6.48Trp, 6.51Phe, 6.52Phe, 3.36Cys, 2.57Val, 5.42Ser, 5.43Ser, 5.46Ser
**SPIPERONE**	3.32Asp, 6.48Trp, 5.42Ser, 5.43Ser, 5.46Ser, 3.36Cys, 6.55His/Asn, 2.57Val
**CHLORPROMAZINE**	3.32Asp, 6.48Trp, 5.42Ser, 5.43Ser, 5.46Ser, 6.55His/Asn, 3.36Cys, 6.51Phe

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
