# Peer review of "A Complete Assessment of Dopamine Receptor- Ligand Interactions through Computational Methods"

_molecules, 2019, doi:10.3390/molecules24071196_

Round 1

Reviewer 1 Report

The work of Buschbell et al. is a very comprehensive work using an in silico approach, to generate dopamine receptor-ligand complexes of the five different dopamine receptors. It is very rich in terms of ligands used and data presented. However, the paper is very difficult to read for a non-specialist of the DR field, as I am. The introduction does not yet give all needed details to enable a reader to understand the paper. For example, the available structural information on these receptors is not detailed at all. It remains unclear for me, why the authors made homology models the receptors for which structural information is available. I think it should be clearly state – maybe already in the introduction. Also the result section and the discussion section are too descriptive and should be shortened significantly. Large parts are repetitive and could be moved to the M&M. Also it would be good sometimes to refer to the M&M in the results sections. For example, the section 3.2 is for me not really a discussion of the results but more an argument why this scoring approach would be used. It could be moved to the M&M section. Furthermore, I suggest a reorganization of the results and the discussion section to make the text easier to read. I would suggest to change the heading and reorganize different sections. For example, for the results section the heading should not be named after the technique used but after the result for example section 2.1 and 2.2 could be easily be merged into a section “Homology 3D Structures of D1R-D5R are stable” Also the heading for 2.3.1 could be part of 2.3 and the subsections could be organized based on the studied receptors. The same is true for the Discussion section.  Also some of the figures contain not all information needed to understand the figure. For example, figure 3 contains colors that are not mentioned in the caption.  In my opinion this manuscript needs significant reworking and would be much easier to read afterwards. Like that the authors could also highlight better their major findings, which might be easily overseen in the current version of the manuscript. I put a few major and minor remarks below which might further help the authors to rework their manuscript. However, as it is a very large and comprehensive work and of high interest for the readers of “Molecules”, I would suggest to  consider it after major revisions.

Major remarks:

·       The abstract is very difficult to follow starting at line 19 (page 1). It would be good to explain a bit more here. What is the DR sub-type specificity, why is the TM1-TM2-TM3-TM7 microdomain so important, what are unspecfic interactions and why larger then 2.5 Angstroem; maybe remove that the approach rebuild known receptor-inhibitor complexes from the conclusion to make more space explaining the results?

·       In the introduction (section 1.1) A lot of different drugs are mentioned here, but it would be nice to make a clearer link between them and the disease they should treat. Would make the text easier to read and more understandable. Maybe also a figure showing the chemical composition of these compounds might be nice.

·       Section 1.1 needs to have a link to the structure of these receptors. It should be summarized what is known about the structural organisation of these proteins. A figure of the general fold (if there is one) would be good as well – maybe with a zoom on the binding site?

·       Page 3 lines 83/84. It should be mentioned in that in the ligand based approach only the structure of the ligand is used and not the one of the protein. Also it might be worth mentioning that we have to know several ligands and their affinities. Maybe an example for both approaches would be good here.

·       Section 1.2: It would be good to mention shortly advantages and disadvantages of each approach.

·       Section 2.1: This section should be better organized. A table would be good the different identity values. Also it remains unclear for me why proteins for which we have Xray structures should be remodelled.  The version of the Modeller program should come in the Material and methods section. The pdb codes are mentioned too often. They should appear in the M&M section.  Also the available crystal structures should have been introduced in the introduction, not here.  

·       P3 l 114-116 “The D2R model was modelled with the crystallographic structure of 114 the D2R complexed with risperidone (PDBid: 6CM4) [29], (total similarity 97.0% with 115 BLAST and 100.0 % with ClustalOmega).” Why is it needed to make a model of the D2R if there is already a PDB file? I am not following here –please elaborate. Same comment for the D3R model and the D4R model. For these structures it is not even explained if these are Xray structures of a complex or of a ligand-free protein

·       Section 2.2 : This section should mention who long the MD simulations where. Also mention the number of repetitions for each MD? Please also state whether the MD simulations have been performed on the ligand-free or ligand-bound protein.  Please add if the RMSD was done on all atoms or only on the Calpha atoms (either here in the text or in the Figure S1)

·       P5. 160 “Overall, the five models showed good overall stability. “ The five models, does that mean the five models of each receptor or did you do only model per receptor? Please clarify

·       Figure S2 defines the used site for the docking. I strongly suggest it to be moved either as an individual figure or as a part of a figure to the main text. Otherwise the docking section is very difficult to understand.

·       P 6. Lines 209 to 218: this part for me it too descriptive and should be moved (partially) to the M&M  section.

·       P. 6 lines 227 following. I think it would be very important to state here more clearly that you are for the moment only considering dopamine. And then in Figure 1 again add dopamine in the figure caption. Also explain why you used dopamine first and not another ligand. Are there any Xray structures with dopamine bound? Would it be possible to compare with them here?

·       Figure 3: Please add a scale next to the plots. There are more colors then red and blue-violet as explained below the figure

·       P 12/13 lines 360 to 368 this is too detailed and should go the M&M

Minor remarks:

·        p 1 l. 14 „evolving these receptors“, either involving these receptors or evolving from these receptors

·       P 1 l.33 It would be nice to say that you are talking about the human dopaminergic system. Or are these system also studied in other organisms

·       P. 2 line 44-45 “The Dopamine 44 Receptors (DR) belong to the G-protein-coupled receptors (GPCRs)” would be good to add here the G-protein-coupled repector family (to make clear that it is a family).

·       P 2 . 47 “a significant target of pharmacotherapeutics.” A reference would be good here.

·       P2. Lines 47 to 49 Would be nice to say for which diseases these mentioned drugs are used

·       P2 lines 50 to 51: Would be nice to add a reason (if known) why these drugs have side effects and nonselective profiles

·       P3 line “The D1R was modelled with the” was modelled based on or was modelled using XX as a template

·       P3. line 122: „TMs in“please introduce the abbreviation TM before using it

·       P 3. Line 121 “The D3R was modelled using 3PBL” please add which protein 3PBL  is (D3R)

·       P4 lines 124-127 “Regarding the D1-like 124 subtypes, the receptors were not modelled with their own crystal structure template 125 since they are not available yet. D1R was modelled with the crystal structure of the 126 D3R (PDBid: 3PBL) whereas D5R was modelled with the crystal structure of the D4R 127 (PDBid: 5WIU).“ Normally modelling is done if the structure is not known. Why mentioning it here? On the top, this information is coming too late.

·       P4 line 145: “Pro-SA and ProQ analysis” Please give a reference to these programs

·       P4 line 151-152 “All final DR models (Table 1Error! 151 Reference source not found.) achieved” Please correct

·       P4 Table 1: not sure I understand the table, is this an average value or is it for the best model, please clarify in the table heading.

·       P5 line 166 and 167 “In this work, we used the comprehensive review of Floresca and Schnetz (2004) 166 [38], highly used [39–41], as a base for the definition of the binding pocket of all 167 dopamine receptors.” This sentence is unclear to me, could you quickly explain what they did? Or simply give their definition of the binding site?

·       P5 “Furthermore, by applying Ballesteros & Weinstein numbering 168 (B&W) [42] the position of considered important residues was more easily 169 comparable between all receptors“ Please explaine quickly the B&W numbering for people which are not used to it (or add it in the M&M section)

·       P6 l.209 “After 100 ns MD simulations of each model, 10 conformational rearrangements 209 plus initial model (time 0 ns) were chosen for each receptor and subjected to 210 molecular docking of 15 different ligands“ Please explain how these conformations were selected (after a certain time?) or did you do clustering on the MD trajectories?

·        P.6 lines 277 following “For a general overview, binding poses with more than 5 conformations per  cluster were considered as a valid ligand position, despite the Binding Energy (BE) of this pose (Figure 1A).” This has to come before the re-docking of the crystallized ligands is decribed.

·       P 10 lines 317 “(Error! Reference source 317 not found.), 3” Please add the correct reference

·       P13 lines 381 “other interactions ranging from 50 to 0 for all ligands at all DRs (Error! Reference 381 source not found.Error! Reference source not found.). In“ please correct

·       P. 24 lines 838 and following “The prepared receptor structures were 838 inserted into a rectangular box simulation with dimensions of 114 x 114 x 107 Å.” The protein was inserted in the membrane, or not?

·       P. 24 line 848: Did you use  CMAP correction for the protein?

Author Response

General comments:

The work of Buschbell et al. is a very comprehensive work using an in-silico approach, to generate dopamine receptor-ligand complexes of the five different dopamine receptors. It is very rich in terms of ligands used and data presented. However, the paper is very difficult to read for a non-specialist of the DR field, as I am.  The introduction does not yet give all needed details to enable a reader to understand the paper. For example, the available structural information on these receptors is not detailed at all. It remains unclear for me, why the authors made homology models the receptors for which structural information is available. I think it should be clearly state – maybe already in the introduction.

GPCRs are the target of a large majority of drugs available in the market and their overall structure is well defined in the field and in a large number of reviews such as: Fredriksson: The GPCRs in the Human Genome Form Five Main Families. Phylogenetic Analysis, Paralogon Groups, and Fingerprints (2003); Rosenbaum et al.: The structure and function of GPCRs (2009); Latek et al.: GPCRs-recent advances (2012). The number of crystal structure available for this huge family of more than 900 receptors has been steady increasingly but in the case of dopamine receptors, the ones in study here, only D2-like (D2R, D3R and D4R) receptors can be found in the Protein Data Bank. However, even those do not have all necessary detail. In fact, to attain a good crystal structure for these receptors, researchers need some engineering such as lysozyme or other proteins addition. This leads to the lack of the intracellular loop3, a crucial point for intracellular partner coupling. As such, all 5 receptors were modeled with the available crystal structures to obtain complete structures for all. We tried to clarify these facts in the revised version of the manuscript. Please check P5, L152).

Also, the result section and the discussion section are too descriptive and should be shortened significantly. Large parts are repetitive and could be moved to the M&M.  Also, it would be good sometimes to refer to the M&M in the results sections. For example, the section 3.2 is for me not really a discussion of the results but more an argument why this scoring approach would be used. It could be moved to the M&M section. Furthermore, I suggest a reorganization of the results and the discussion section to make the text easier to read. I would suggest changing the heading and reorganize different sections. For example, for the results section the heading should not be named after the technique used but after the result for example section 2.1 and 2.2 could be easily be merged into a section “Homology 3D Structures of D1R-D5R are stable” Also the heading for 2.3.1 could be part of 2.3 and the subsections could be organized based on the studied receptors. The same is true for the Discussion section. Also, some of the figures contain not all information needed to understand the figure. For example, figure 3 contains colors that are not mentioned in the caption.  In my opinion this manuscript needs significant reworking and would be much easier to read afterwards. Like that the authors could also highlight better their major findings, which might be easily overseen in the current version of the manuscript. I put a few major and minor remarks below which might further help the authors to rework their manuscript. However, as it is a very large and comprehensive work and of high interest for the readers of “Molecules”, I would suggest to consider it after major revisions.

We greatly appreciate the reviewer’s suggestions and advices. We shorted results and discussion sections in order to make it clearer to the reader, parts were moved to M&M (mentioned when needed) and sections names were changed. Please check revised manuscript.

Major remarks

1.     The abstract is very difficult to follow starting at line 19 (page 1). It would be good to explain a bit more here. What is the DR sub-type specificity, why is the TM1-TM2-TM3-TM7 microdomain so important, what are unspecific interactions and why larger then 2.5 Angstrom; maybe remove that the approach rebuild known receptor-inhibitor complexes from the conclusion to make more space explaining the results?

We reorganized results at the abstract section to be clearer and possess a lower dependence of knowledge gathered by reading in detail the manuscript. Please check P1 of the revised manuscript.

2.     In the introduction (section 1.1) A lot of different drugs are mentioned here, but it would be nice to make a clearer link between them and the disease they should treat. Would make the text easier to read and more understandable. Maybe also a figure showing the chemical composition of these compounds might be nice.

We added Figure 1 (P3, L82) and clarified their role in section 1.1. In addition, we rearranged the introduction to make it more structured and easier to follow.

3.     Section 1.1 needs to have a link to the structure of these receptors. It should be summarized what is known about the structural organization of these proteins. A figure of the general fold (if there is one) would be good as well – maybe with a zoom on the binding site?

We followed reviewer’s suggestion and added a new phrase that clarifies the GPCR overall fold. Please check P2, L50-54 of the revised manuscript.

4.     Page 3 lines 83/84. It should be mentioned in that in the ligand-based approach only the structure of the ligand is used and not the one of the protein. Also, it might be worth mentioning that we have to know several ligands and their affinities. Maybe an example for both approaches would be good here.

We followed reviewer’s suggestion and added a new paragraph at P4, L107-116 of the revised manuscript.

5.     Section 1.2: It would be good to mention shortly advantages and disadvantages of each approach.

We followed reviewer’s suggestion and added a new paragraph at P4, L116-120 of the revised manuscript.

6.     Section 2.1: This section should be better organized. A table would be good the different identity values. Also, it remains unclear for me why proteins for which we have Xray structures should be remodelled. The version of the Modeller program should come in the Material and methods section. The pdb codes are mentioned too often. They should appear in the M&M section.  Also, the available crystal structures should have been introduced in the introduction, not here.

We reorganized section 2.1. and moved parts to the M&M section such as the PDB codes and the version of MODELLER. Please check pages 5 and 21 of the revised manuscript. We also listed the DR crystal structures already in the Introduction section (P4, L131-135), and better explained the modeling templates (P5, L152-161).

7.     P3 l 114-116 “The D2R model was modelled with the crystallographic structure of 114 the D2R complexed with risperidone (PDBid: 6CM4) [29], (total similarity 97.0% with 115 BLAST and 100.0 % with ClustalOmega).” Why is it needed to make a model of the D2R if there is already a PDB file? I am not following here –please elaborate. Same comment for the D3R model and the D4R model. For these structures it is not even explained if these are Xray structures of a complex or of a ligand-free protein.

We added the new paragraph at section 2.1 to better clarify this question: “The ligand-free D2-like homology models were generated using the resolved ligand-bounded crystal structures of the D2R (PDBid:  6CM4) [38], D3R (PDBid: 3PBL) [39] and D4R (PDBid: 5WIU) [42] (over 90 % identity). 3D crystal structure of DRs are typically incomplete lacking key regions for intracellular partner coupling such as IntraCellular Loop 3 (ICL3). In contracts, D1-like DRs lack their own templates and therefore the most suitable template to each DR was selected according to the percentage of similarity obtained upon sequence alignment by BLAST [46] in combination with ClustalOmega [47]. In fact, the D3R crystal structure was chosen as template for D1R (35.0 % identity with BLAST and 39.5 % with ClustalOmega), and D5R models (total similarity of 35.0 % BLAST/ 39.1 % ClustalOmega, Table 2)”. Please check P5, L152-163 of the revised manuscript.

8.     Section 2.2 : This section should mention how long the MD simulations where. Also mention the number of repetitions for each MD? Please also state whether the MD simulations have been performed on the ligand-free or ligand-bound protein.  Please add if the RMSD was done on all atoms or only on the Calpha atoms (either here in the text or in the Figure S1)

We performed one replica of each ligand-free modeled receptor. The RMSD was calculated only for C-alphas. We clarified this information in P5, L182-184 of the revised manuscript.  

9.     P5. 160 “Overall, the five models showed good overall stability. “The five models, does that mean the five models of each receptor or did you do only model per receptor? Please clarify.

We assigned our best and single model for molecular docking for each receptor and subjected it to MD simulation to attain different possible conformations. We clarified this information by adding “Each DR model showed good overall stability.” P6, L184-184 of the revised manuscript.

10.  Figure S2 defines the used site for the docking. I strongly suggest it to be moved either as an individual figure or as a part of a figure to the main text. Otherwise the docking section is very difficult to understand.

We followed reviewer’s suggestion and added this figure to the main text. Please check new Figure 2, P6.

11.  P 6. Lines 209 to 218: this part for me it too descriptive and should be moved (partially) to the M&M section.

We moved parts of the 2.1 section to the corresponding M&M section. Please check pages 7 and 22 of the revised manuscript.

12.  P. 6 lines 227 following. I think it would be very important to state here more clearly that you are for the moment only considering dopamine. And then in Figure 1 again add dopamine in the figure caption. Also explain why you used dopamine first and not another ligand. Are there any Xray structures with dopamine bound? Would it be possible to compare with them here?

The dopamine-bounded DR crystal structure cannot be found in the PDB as dopamine’s structural properties are not suitable for crystallization (too small, not suitable for stabilizing a GPCR). However, Floresca&Schetz produced a large mutagenesis study/review about the binding mechanism of dopamine, which usually guides in silico predictions at this type of GPCRs. We clarified in Figure 3 and P8, L262-267 of the revised manuscript.

13.  Figure 3: Please add a scale next to the plots. There are more colors then red and blue-violet as explained below the figure.

We clarified the continuous range of colors in this figure. Please check P13 of the revised manuscript.

14.  P 12/13 lines 360 to 368 this is too detailed and should go the M&M.

We followed reviewer’s advice and changed section to M&M.

Minor remarks:

1.     p 1 l. 14 „evolving these receptors“, either involving these receptors or evolving from these receptors

We followed the suggestion of reviewer 1 and corrected the expression to “involving these receptors”. Please check P1, L16 of the revised manuscript.

2.     P 1 l.33 It would be nice to say that you are talking about the human dopaminergic system. Or are these system also studied in other organisms

We added the following sentence at introduction to clarify that the mammalian dopaminergic receptors were the studied system herein: “The importance of dopamine has dramatically emerged from being just an intermediary in the formation of noradrenaline to having a celebrity-status as the most important mammalian neurotransmitter.” Please check P2, L42-44 of the revised manuscript.

3.     P. 2 line 44-45 “The Dopamine 44 Receptors (DR) belong to the G-protein-coupled receptors (GPCRs)” would be good to add here the G-protein-coupled receptor family (to make clear that it is a family).

We added the following expression to clarify that the GPCRs are a protein family: “The DRs belong to the G-Protein-Coupled Receptor family (GPCRs), the largest and most diverse protein family in humans with approximately 800 members [5,6]. Please check P2, L50-52 of the revised manuscript.

4.     P. 2 line 47 “a significant target of pharmacotherapeutics.” A reference would be good here.

We added reference to following expression: significant target of pharmacotherapeutics [6] -> Thimm, D.; Funke, M.; Meyer, A.; Müller, C. E. 6-Bromo-8-(4- methoxybenzamido)-4-oxo-4 H-chromene-2-carboxylic Acid: A powerful tool for studying orphan G protein-coupled receptor GPR35. J. Med. Chem. 2013, 56 (17), 7084–7099. Please check P2, L55 of the revised manuscript.

5.     P2. Lines 47 to 49 Would be nice to say for which diseases these mentioned drugs are used

We followed reviewer’s suggestion and added a new figure (Figure 1) mentioning some of the various conditions that are currently treated with theses antipsychotics. Please check P3 of the revised manuscript.

6.     P2 lines 50 to 51: Would be nice to add a reason (if known) why these drugs have side effects and nonselective profiles.

The reviewer asked for reasons why antipsychotics have side effects and non-selective profiles. Therefore, we added the following lines: “It was then later discovered that these multiple clinical and adverse effects of several antipsychotics depended on the combination of occupied receptors from other systems such as cholinergic, histaminergic and serotoninergic receptors (but always including the D2R) resulting in non-selective profiles and therefore in an insufficient explanation of the mechanism of action [11,15]”.  Please check P2, L69-74 of the revised manuscript.

7.     P3 line “The D1R was modelled with the” was modelled based on or was modelled using XX as a template.

We changed the expression “The D1R was modelled with the” to “In fact, the D3R crystal structure was chosen as template for D1R”. Please check P5, L159-160 of the revised manuscript.

8.     P3. line 122: „TMs in“please introduce the abbreviation TM before using it.

We introduced the abbreviation “TMs” on P2, L53 of the revised manuscript.

9.     P 3. Line 121 “The D3R was modelled using 3PBL” please add which protein 3PBL is (D3R)

We clarified this point in the revised version. Please check P5, line 36.

10.  P4 lines 124-127 “Regarding the D1-like 124 subtypes, the receptors were not modelled with their own crystal structure template 125 since they are not available yet. D1R was modelled with the crystal structure of the 126 D3R (PDBid: 3PBL) whereas D5R was modelled with the crystal structure of the D4R 127 (PDBid: 5WIU).“ Normally modelling is done if the structure is not known. Why mentioning it here? On the top, this information is coming too late.

We reorganized the first paragraph of the results section in order to clarify the homology modeling done herein. Please check 2.1.P5, L152-163.

11.  P4 line 145: “Pro-SA and ProQ analysis” Please give a reference to these programs.

We added references to both “Pro-SA and ProQ” programs. Please check P5 L177-178 of the revised manuscript.

12.  P4 line 151-152 “All final DR models (Table 1Error! 151 Reference source not found.) achieved” Please correct

We corrected the source of Table 1. Please check P5, L196.

13.  P4 Table 1: not sure I understand the table, is this an average value or is it for the best model, please clarify in the table heading.

We changed this table title in order to clarify that the values listed here are the absolute values for the best/final model. Please check P5, L164 of the revised manuscript.

14.  P5 line 166 and 167 “In this work, we used the comprehensive review of Floresca and Schnetz (2004) 166 [38], highly used [39–41], as a base for the definition of the binding pocket of all 167 dopamine receptors.” This sentence is unclear to me, could you quickly explain what they did? Or simply give their definition of the binding site?

Floresca&Schnetz is a review about mutagenesis studies and testing of known dopaminergic ligands from which they depicted some amino-acids in the DRs binding pockets. We added Figure 2 to better clarify the initial overall residues used for docking of the various ligands.  Please check P6, L193-195 of the revised manuscript.

15.  P5 “Furthermore, by applying Ballesteros & Weinstein numbering 168 (B&W) [42] the position of considered important residues was more easily 169 comparable between all receptors“ Please explaine quickly the B&W numbering for people which are not used to it (or add it in the M&M section)

As suggested by reviewer 1 we added an explanation to Ballesteros&Weinstein nomenclature (B&W). Please check P6, L197-200 of the revised manuscript.

16.  P.6 l.209 “After 100 ns MD simulations of each model, 10 conformational rearrangements 209 plus initial model (time 0 ns) were chosen for each receptor and subjected to 210 molecular docking of 15 different ligands“ Please explain how these conformations were selected (after a certain time?) or did you do clustering on the MD trajectories?

The conformations were retrieved at every 5 ns for the last 50 ns of the MD trajectory upon attaining stabilization. This is better explained at the M&M section. Please check pages 7 and 24 of the revised manuscript.

17.  P.6 lines 277 following “For a general overview, binding poses with more than 5 conformations per cluster were considered as a valid ligand position, despite the Binding Energy (BE) of this pose (Figure 1A).” This has to come before the re-docking of the crystallized ligands is decribed.

As suggest by the reviewer, we moved the sentence “For instance, binding poses with more than 5 conformations per cluster were considered as a valid ligand position, despite the Binding Energy (BE) of this pose” before the sections about the redocking of the crystalized structures. Please check P7, L250-252 of the revised manuscript.

18.  P 10 lines 317 “(Error! Reference source 317 not found.), 3” Please add the correct reference

We corrected this reference. Please check P8, L262 of the revised manuscript.

19.  P13 lines 381 “other interactions ranging from 50 to 0 for all ligands at all DRs (Error! Reference 381 source not found.Error! Reference source not found.). In“ please correct

We corrected this reference. Please check P8, L272 of the revised manuscript.

20.  P. 24 lines 838 and following “The prepared receptor structures were 838 inserted into a rectangular box simulation with dimensions of 114 x 114 x 107 Å.” The protein was inserted in the membrane, or not?

We clarified our explanation in the M&M sections as follows: “The prepared receptor structures were inserted into a previously constructed lipid bilayer of POPC: Cholesterol (9:1). Insertion of the receptors in the membrane was performed with g_membed package of GROMACS [99]. System was then solvated with explicitly represented water. Sodium and chloride ions were added to neutralize the system until it reached a total concentration of 0.15 M. The final systems dimensions were 114 x 114 x 107 Å and included approximately 370 POPC, 40 cholesterols, 125 sodium ions, 139 chloride ions and 28500 water molecules, with small variations from receptor to receptor.” Please check P23, L722-729 of the revised manuscript.

21.  P. 24 line 848: Did you use CMAP correction for the protein?

CHARMM36 already has implemented CMPA correction and also improved CMAP corrections from the previous version (CHARMM22/CMAP). We clarified in the revised manuscript. Please check P23, L731.

Reviewer 2 Report

The manuscript by Itina S. Moreira et al. describes the full assessment of known dopamine receptor^ligand interaction with MD, BINANA argolism and so on.The manuscript is written clearly, structured well und the experiments seem to be performed accurate. It is surely in the scope of the journal and will be of interest for the readership. Nevertheless, some minor revisions will have to be performed to make it suitable for publication in "Molecules".

1, The reason to choose the BINANA for the studies still unclear, because BINANA demonstrated the assessment of the interaction of non-GPCR family. More explanation should be needed for readers.

2, on line 381-382 the reference should be corrected.

Author Response

The manuscript by Itina S. Moreira et al. describes the full assessment of known dopamine receptor-ligand interaction with MD, BINANA argolism and so on. The manuscript is written clearly, structured well und the experiments seem to be performed accurate. It is surely in the scope of the journal and will be of interest for the readership. Nevertheless, some minor revisions will have to be performed to make it suitable for publication in "Molecules".

1.     The reason to choose the BINANA for the studies still unclear, because BINANA demonstrated the assessment of the interaction of non-GPCR family. More explanation should be needed for readers.

WE followed reviewer’s advice and added the following phrase to introduction to clarify BINANA’s role in this study: “BINANA was shown to successfully atomically characterize key interactions between protein amino-acids and ligand atoms, and as such it is a promising approach to map in detail such interactions in GPCRs.” Please check P4-5, L143-145 of the revised manuscript.

2.     On line 381-382 the reference should be corrected.

Reference was added to P14, L395 of the revised manuscript.

Reviewer 3 Report

The authors reported an extensive in silico evaluation of the ligand-dopamine receptors interactions. I suggest to publish it in a journal focused on computational methodologies.

In my view, the present form of the paper cannot be accepted for publication as an article in Molecules for the following reasons:

·       The computational study has been carried out with technical diligence; however, the obtained results will not be useful for scientists involved in the dopamine filed. The obtained data have been reported without a proper analysis; they have been reported as a list of information without an analysis and a summary of the main important features.

·       The relevance of this paper is focused on the quality and reliability of the applied computational approach. However, the authors applied the computational procedure without an experimental evaluation. The authors should experimentally test new hits designed/screened on the basis of their computational studies verifying the obtained results

 Minor:

Along the text there are errors correlated with missing reference links (for example, page 4 line 151)

Author Response

CADD approaches are known to be highly relevant and used in pharmaceutical industries such as Pfizer, GlaxoSmithKline and Roche (e.g. GSK mentioned in an article by Warr. A CADD-a log of strategies in pharma, 2017).

We have clarified paper’s structure in order to better show our main structural findings that will help future researchers design more specific drugs towards dopamine receptors. E.g. that the aromatic microdomain is responsible for the majority of ligand interactions to all DRs;  hydrophobic interactions form a huge network across TM2, TM3, TM7, salt-bridges are formed conservative with 3.32Asp, however for some cases amino acids on ECL1 were highlighted also as relevant; “bulky” ligands like bromocriptine and clozapine were not able to form salt-bridges; for D5R, an HB between 5.38Tyr and dopamine was stressed out as unique for all ligands; TM1 and TM2 were shown to be relevant for binding large ligands and lastly that especially the formation of electrostatic interactions with phenylalanine, tyrosine and tryptophan may be a key to selectivity regarding the D1-like DR.

1.     Minor remarks: Along the text there are errors correlated with missing reference links (for example, page 4 line 151)

We corrected the reference of Table 1. P5, L163 (also listed by reviewer 1, now moved to the Materials and Methods section)

Round 2

Reviewer 1 Report

The work of Buschbell et al. is a very comprehensive work using an in silico approach, to generate dopamine receptor-ligand complexes of the five different dopamine receptors.  The first version of the manuscript was very difficult to read. In my opinion the current version of the manuscript is largely improved over the previous version and is much easier to read. The readability of the manuscript could be still improved by adding for example the table of all ligand used in the beginning of the manuscript and the text could be better linked to the figures shown within the text. Since I still have some minor remarks, which are summarized below, and I recommend minor revision.

·       p.1 line 26: would be good to introduce TM before the abbreviation is used

·       p.3 lines 86 to 97: It would be good to show the discussed inhibitors in the Figure 1

·       p.4 lines 107 to 120: I would suggest to reorder the paragraph. First discuss the different approaches and afterwards give the examples

·       p.5 lines 177 to 184: The RSMD was computed for the overall protein. Might be interesting to check it for the different TMs as well

·       p.5 line 183: Do you mean by homology scores sequence identity?

·       P.6 lines 192 to 198 It would be nice to add both binding sites in the figure (the OBP and SBP)

·       P.6 line 218: It is not clear which residues are chosen flexible in the docking, maybe a link with the M&M section would be nice?

·       P.7 lines 231 to 232: It would be nice to give reference(s) to the sentence with the experiments done on the SBP

·       P7 line 234 to 235: I think the sentence referring to Table S2 should come earlier in the paragraph

·       P8: figure 3: the curve in 3A is very small, maybe the scale on the Y axis could be adapted? The figure 3B is very difficult to interpret. I would suggest to only show the cluster with the highest number of points and not all of them. The number of the used cluster could be added to the figure or the description of the figure

·       P9: section 2.4 A table of all the used molecules would be good to have here to better understand the text (maybe move table 3 to here). This section is also a bit difficult to read. Maybe it would be easier to understand if first for all docked molecules first the energy and then the NoC is discussed? Also it remains unclear to me why there are only 7 molecules discussed but 15 were docked. Maybe explain better

·       P10: Figure 4: difficult to interprete see remark to figure 3

·       P12: figure 5: I think blank is not what is seen, what is measured is that the distance is > 12 Angstroem

·       P13 line 379: I am not sure to understand what are interactions at 2.5 and 4. Do you mean between 2.5 and 4 and higher than 4? Or do you mean below 2.5 and afterwards between 2.5 and 4? Maybe clarify

·       P16: Figure 6: It would be good to know what were the numbers of frames used for the plot. What does an interaction of 100 mean? Does that correspond to 100% of the trajectories? Or maybe give the interactions in percentages?

·       P23 table 3: It would be interesting to add here a short description of the OBP and the SBP

Author Response

REVIEWER 1

General Comments:

The work of Buschbell et al. is a very comprehensive work using an in silico approach, to generate dopamine receptor-ligand complexes of the five different dopamine receptors.  The first version of the manuscript was very difficult to read. In my opinion the current version of the manuscript is largely improved over the previous version and is much easier to read. The readability of the manuscript could be still improved by adding for example the table of all ligand used in the beginning of the manuscript and the text could be better linked to the figures shown within the text. Since I still have some minor remarks, which are summarized below, and I recommend minor revision.

Minor revisions:

1.     p.1 line 26: would be good to introduce TM before the abbreviation is used

We followed the reviewer’s suggestion and introduced TM before the abbreviation at P1, L 26-27.

2.     p.3 lines 86 to 97: It would be good to show the discussed inhibitors in the Figure 1

We followed the reviewer’s suggestion and added the examples of the DR-selective ligands (not inhibitors) to Figure 1 (P3, L84-86).

3.     p.4 lines 107 to 120: I would suggest to reorder the paragraph. First discuss the different approaches and afterwards give the examples

The paragraph was reordered as suggested by the reviewer. Please check P4, L107-122.

4.     p.5 lines 177 to 184: The RSMD was computed for the overall protein. Might be interesting to check it for the different TMs as well

The RMSD values for the different TMs were computed and are listed in a new SI table, Table S2. A link to this table was also added in the main text. Please check P6, L189-191.

5.     p.5 line 183: Do you mean by homology scores sequence identity?

 Yes. It is shown in the first paragraph of section 2.1 that D2-like receptors have higher sequence identity. The higher the identity, the better and more stable the model. This is in accordance with the values of RMSD showed in P6, L184-189

6.     P.6 lines 192 to 198 It would be nice to add both binding sites in the figure (the OBP and SBP)

We colored the areas of the orthosteric and secondary binding pocket in Figure 1. Please check P6, L202.

7.     P.6 line 218: It is not clear which residues are chosen flexible in the docking, maybe a link with the M&M section would be nice?

A link to Table 4 of M&M section was added at P7, L245-247.

8.     P.7 lines 231 to 232: It would be nice to give reference(s) to the sentence with the experiments done on the SBP

We followed the reviewer’s suggestion and added the experiments performed in these reference studies. We added following term:  “Different authors hypothesized that DRs have a secondary binding pocket (SBP) next to the OBP, which was confirmed by the resolved crystal structures together with computational analyses [37,38,59]”. Please see P7, L272-274.

9.     P7 line 234 to 235: I think the sentence referring to Table S2 should come earlier in the paragraph.

The sentence referring to Table S2 (new Table S3) was moved up in the paragraph. Please check P7, L280-282.

10.  P8: figure 3: the curve in 3A is very small, maybe the scale on the Y axis could be adapted? The figure 3B is very difficult to interpret. I would suggest to only show the cluster with the highest number of points and not all of them. The number of the used cluster could be added to the figure or the description of the figure.

The scale in figure 3A was adapted. For figure 3B we calculated the mean of the three best clusters given by AutoDock4.2. Please check new Figure 3, P9.

11.  P9: section 2.4 A table of all the used molecules would be good to have here to better understand the text (maybe move table 3 to here). This section is also a bit difficult to read. Maybe it would be easier to understand if first for all docked molecules first the energy and then the NoC is discussed? Also it remains unclear to me why there are only 7 molecules discussed but 15 were docked. Maybe explain better

We moved table 3 to section 2.4. Please check P10, 359. We changed the BE and Noc part in section 2.4 and reorganized the text. Please see P11-12, 362-382.

We focussed on the selection of these 7 ligands, as example of ligands with different DR selectivity.  Detailed information on all 15 ligands in two graphs would be difficult to list. Nevertheless, everything can be found on Tables S4-8 and Figure S7

12.  P10: Figure 4: difficult to interprete see remark to figure 3

The scale of the binding energies in figure 4 was adapted. For the number of conformations we calculated the mean of the three best clusters given by Autodock4.2. Please check figure 4 on P13, L347.

13.  P12: figure 5: I think blank is not what is seen, what is measured is that the distance is > 12 Angstroem

The white color indicates distances between 11A and 12A. Higher values were not attained here, and the grey/blank cells are also marked with a cross. For these ligand : residue-pairs no distances were measured since these combinations did not appear in the docking decoys. Please see also the new comment in figure 5: “Noteworthy is that not all ligands were set to interact with all residues shown in the x-axis in the molecular docking. (e.g. only clozapine and aripiprazole were set to interact with 3.33Val).”, P15, L388-389.

14.  P13 line 379: I am not sure to understand what are interactions at 2.5 and 4. Do you mean between 2.5 and 4 and higher than 4? Or do you mean below 2.5 and afterwards between 2.5 and 4? Maybe clarify

Interactions were measured below or equal to 2.5A and below or equal 4A. Therefore, the interactions below or equal 4A also include the 2.5A interactions. We added a clarification at P15-16, L396-397.

15.  P16: Figure 6: It would be good to know what were the numbers of frames used for the plot. What does an interaction of 100 mean? Does that correspond to 100% of the trajectories? Or maybe give the interactions in percentages?

The values in Figure 6 correspond to a summarized data at all timepoints for each ligand. The timepoints used, as described in M&M, were the initial structure and the snapshots retrieved every 5ns after the first 50 ns. The scale is in “number of interactions”. For example, colored as >100 are listed the ones with more than 100 interactions of that kind.

16.  P23 table 3: It would be interesting to add here a short description of the OBP and the SBP

Since following reviewer suggestion of moving Table 3 to section 2.4, immediately after section 2.2 where OBP and SBP are explained in detail and shown in figure 2, we think that now it will not be necessary to repeat this information on P10.

Reviewer 3 Report

CADD techniques are indeed very useful approaches; however, they need a proper experimental evaluation. As reported in the previous comments the relevance of this paper is focused on the quality and reliability of the applied computational approach. However, the authors applied the computational procedure without an experimental evaluation.

For the acceptance of the manuscript, in my opinion the authors should experimentally test new hits designed/screened on the basis of their computational studies verifying the obtained results.

Author Response

We understand reviewers' criticism but the methodological approaches applied herein are already well established in the field leading to meaningful and useful conclusions. We believe that they are useful for the GPCR research field.